# Improving the accuracy of single-trial fMRI response estimates using GLMsingle

Jacob S Prince[1]*, Ian Charest[2,3], Jan W Kurzawski[4], John A Pyles[5], Michael J Tarr[6], Kendrick N Kay[7]

[1]Department of Psychology, Harvard University, Cambridge, United States; [2]Center for Human Brain Health, School of Psychology, University of Birmingham, Birmingham, United Kingdom; [3]cerebrUM, Département de Psychologie, Université de Montréal, Montréal, Canada; [4]Department of Psychology, New York University, New York, United States; [5]Center for Human Neuroscience, Department of Psychology, University of Washington, Seattle, United States; [6]Department of Psychology, Neuroscience Institute, Carnegie Mellon University, Pittsburgh, United States; [7]Center for Magnetic Resonance Research (CMRR), Department of Radiology, University of Minnesota, Minneapolis, United States

*For correspondence: jacob.samuel.prince@gmail.com

Competing interest: The authors declare that no competing interests exist.

**Abstract** Advances in artificial intelligence have inspired a paradigm shift in human neuroscience, yielding large-scale functional magnetic resonance imaging (fMRI) datasets that provide high-resolution brain responses to thousands of naturalistic visual stimuli. Because such experiments necessarily involve brief stimulus durations and few repetitions of each stimulus, achieving sufficient signal-to-noise ratio can be a major challenge. We address this challenge by introducing *GLMsingle*, a scalable, user-friendly toolbox available in MATLAB and Python that enables accurate estimation of single-trial fMRI responses (glmsingle.org). Requiring only fMRI time-series data and a design matrix as inputs, GLMsingle integrates three techniques for improving the accuracy of trial-wise general linear model (GLM) beta estimates. First, for each voxel, a custom hemodynamic response function (HRF) is identified from a library of candidate functions. Second, cross-validation is used to derive a set of noise regressors from voxels unrelated to the experiment. Third, to improve the stability of beta estimates for closely spaced trials, betas are regularized on a voxel-wise basis using ridge regression. Applying GLMsingle to the Natural Scenes Dataset and BOLD5000, we find that GLMsingle substantially improves the reliability of beta estimates across visually-responsive cortex in all subjects. Comparable improvements in reliability are also observed in a smaller-scale auditory dataset from the StudyForrest experiment. These improvements translate into tangible benefits for higher-level analyses relevant to systems and cognitive neuroscience. We demonstrate that GLMsingle: (i) helps decorrelate response estimates between trials nearby in time; (ii) enhances representational similarity between subjects within and across datasets; and (iii) boosts one-versus-many decoding of visual stimuli. GLMsingle is a publicly available tool that can significantly improve the quality of past, present, and future neuroimaging datasets sampling brain activity across many experimental conditions.

## Editor's evaluation

This important work provides the field of human neuroimaging with a new method to estimate single-trial fMRI responses. The authors provide compelling evidence that their GLMsingle method goes beyond the current state of the art and leads to more reliable estimates. Therefore, this tool will be of interest to researchers using human neuroimaging to study neural responses in condition-rich designs, as is increasingly common in cognitive neuroscience experiments.

## Introduction

Across many scientific disciplines, datasets are rapidly increasing in size and scope. These resources have kickstarted a new era of data-driven scientific discovery (*Richards et al., 2019*; *Jumper et al., 2021*; *Iten et al., 2020*; *Ravuri et al., 2021*; *Schawinski et al., 2018*; *D'Isanto and Polsterer, 2018*). In visual neuroscience, recent efforts to sample individual brains at unprecedented scale and depth have yielded high-resolution functional magnetic resonance imaging (fMRI) datasets in which subjects view thousands of distinct images over several dozen hours of scanning (see *Naselaris et al., 2021* for a review). These exciting "condition-rich" datasets are large enough to propel the development of computational models of how humans process complex naturalistic stimuli. For example, resources such as the Natural Scenes Dataset (NSD, *Allen et al., 2022*), BOLD5000 (*Chang et al., 2019*), and THINGS (*Hebart et al., 2019*) may be useful for advancing our ability to characterize the tuning (*Bao et al., 2020*; *Li and Bonner, 2022*; *Long et al., 2018*; *Kriegeskorte and Wei, 2021*; *Popham et al., 2021*), topography (*Blauch et al., 2022*; *Doshi and Konkle, 2021*; *Zhang et al., 2021*; *Lee et al., 2020*), and computations (*Yamins et al., 2014*; *DiCarlo et al., 2012*; *Freeman et al., 2013*; *Marques et al., 2021*; *Horikawa and Kamitani, 2017*) performed in visual cortex.

The potential of large-scale datasets to reveal general principles of neural function depends critically on signal-to-noise ratio (SNR), which refers to one's ability to reliably measure distinct neural signatures associated with different stimuli or experimental conditions. Diverse sources of noise affect fMRI data, and these noise sources limit the robustness and interpretability of data analyses (*Liu, 2016*; *Kay et al., 2013*). For example, subject head motion, scanner instabilities, physiological noise, and thermal noise all contribute unwanted variability to fMRI data. Noise is especially problematic in studies that sample a large number of conditions, since the number of repetitions of each condition is typically limited, resulting in noisy responses even after trial-averaging.

The approach we have developed to mitigate the effects of noise comes in the context of general linear model (GLM) analysis of fMRI time-series data (*Dale, 1999*; *Monti, 2011*). We assume that the goal of the GLM analysis is to estimate beta weights representing the blood oxygenation level dependent (BOLD) response amplitude evoked by different experimental conditions. In this context,

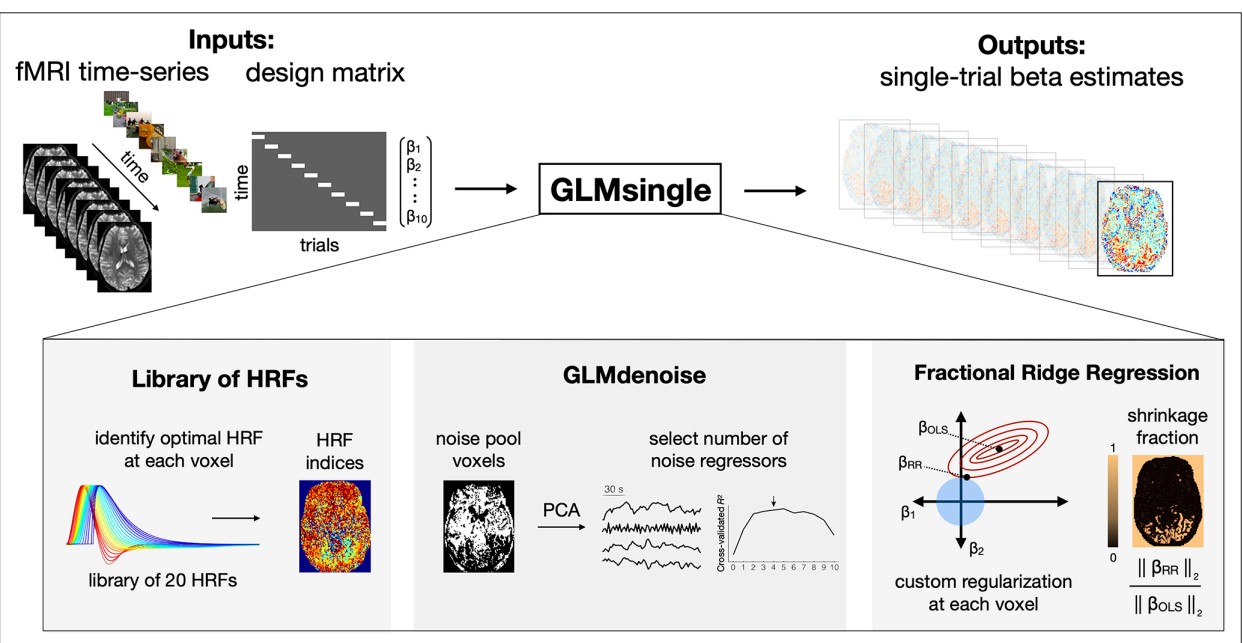

**Figure 1.** Overview of GLMsingle. GLMsingle takes as input a design matrix (where each column indicates the onset times for a given condition) and fMRI time-series in either volumetric or surface space, and returns as output an estimate of single-trial BOLD response amplitudes (beta weights). GLMsingle incorporates three techniques designed to optimize the quality of beta estimates: first, the use of a library of hemodynamic response functions (HRFs), where the best-fitting HRF from the library is chosen for each voxel; second, an adaptation of GLMdenoise (*Kay et al., 2013*) to the single-trial GLM framework, where data-derived nuisance regressors are identified and used to remove noise from beta estimates; and third, an efficient re-parameterization of ridge regression (*Rokem and Kay, 2020*) as a method for dampening the noise inflation caused by correlated single-trial GLM predictors.

we define *noise* as variability observed across repeated instances of a given condition. Therefore, methods that decrease such variability are desirable. Our approach seeks to maximize data quality at the level of individual voxels in individual subjects (as opposed to data quality assessed only at the region or group level), and seeks to obtain response estimates for single trials. These desiderata are powerful; if achieved, they can flexibly support a wide range of subsequent analyses including relating brain responses to trial-wise behavioral measures and pooling data across trials, brain regions, and/ or subjects.

To realize these goals, we introduce *GLMsingle*, a user-friendly software toolbox (with both MATLAB and Python implementations) that performs single-trial BOLD response estimation. Given fMRI time-series data and a design matrix indicating the onsets of experimental conditions, GLMsingle implements a set of optimizations that target three aspects of the GLM framework (*Figure 1*):

1. The choice of hemodynamic response function (HRF) to convolve with the design matrix
2. The inclusion of nuisance regressors that account for components of the data that are thought to be noise
3. The use of regularization to improve the accuracy of the final beta estimates

Importantly, to enable fluid application to even the largest fMRI datasets, GLMsingle is fully automated (no manual setting of parameters) and can be executed efficiently even when gigabytes of fMRI data are passed as input.

We previously used the GLMsingle algorithm to estimate BOLD responses in the NSD dataset (*Allen et al., 2022*). While the optimizations implemented in GLMsingle had a positive impact on data quality, it was not apparent whether the improvements would generalize to other datasets. The goal of this paper is to provide a standalone description of GLMsingle and to rigorously assess performance not only on NSD, but also on BOLD5000 (*Chang et al., 2019*), a distinct large-scale fMRI dataset acquired with different subjects, at different field strength, and with a different experimental design (see Materials and methods). In both datasets, we show that the optimizations implemented in GLMsingle dramatically improve the reliability of GLM beta estimates. We provide further evidence of the general utility of GLMsingle by also evaluating its performance on the music-listening experiment from StudyForrest (*Hanke et al., 2015*). This dataset differs in a number of respects from NSD and BOLD5000: it reflects a non-visual modality, and contains fewer experimental conditions, longer condition durations, a greater number of repetitions per condition, and a jittered inter-stimulus interval. The performance improvements found in this third dataset suggest that GLMsingle may be applicable to a wide range of fMRI tasks and study designs.

We also study the effect of these optimizations on downstream analyses that are of particular relevance to systems and cognitive neuroscience, including representational similarity analysis (RSA; *Kriegeskorte et al., 2008*) and multivoxel pattern analysis (MVPA; *Haxby et al., 2001*; *Norman et al., 2006*; *Poldrack et al., 2011*). In all analyses, we observe improvements in key outcome metrics, suggesting that GLMsingle meaningfully improves the ability of researchers to gain insight into neural representation and computation. Our findings demonstrate that GLMsingle affords the neuroimaging community a clear opportunity for improved data quality. Online materials (code, documentation, example scripts) pertaining to GLMsingle are available at glmsingle.org.

## Results

To assess the impact of GLMsingle, we evaluate four different types of single-trial response estimates (henceforth, *beta versions*). The first arises from a baseline procedure that reflects a typical GLM approach for fMRI analysis (beta version $b1$), and each subsequent beta version ($b2$-$b4$) incorporates an additional strategy for optimizing model fits and mitigating the effects of noise. The final beta version ($b4$) contains the complete set of optimizations provided by the GLMsingle toolbox. The GLMsingle algorithm consists of the following steps:

1. A baseline single-trial GLM is used to model each stimulus trial separately using a canonical HRF. This provides a useful baseline for comparison ($b1$: AssumeHRF).
2. An optimal HRF is identified for each voxel (*Allen et al., 2022*) by iteratively fitting a set of GLMs, each time using a different HRF from a library of 20 HRFs. For each voxel, we identify the

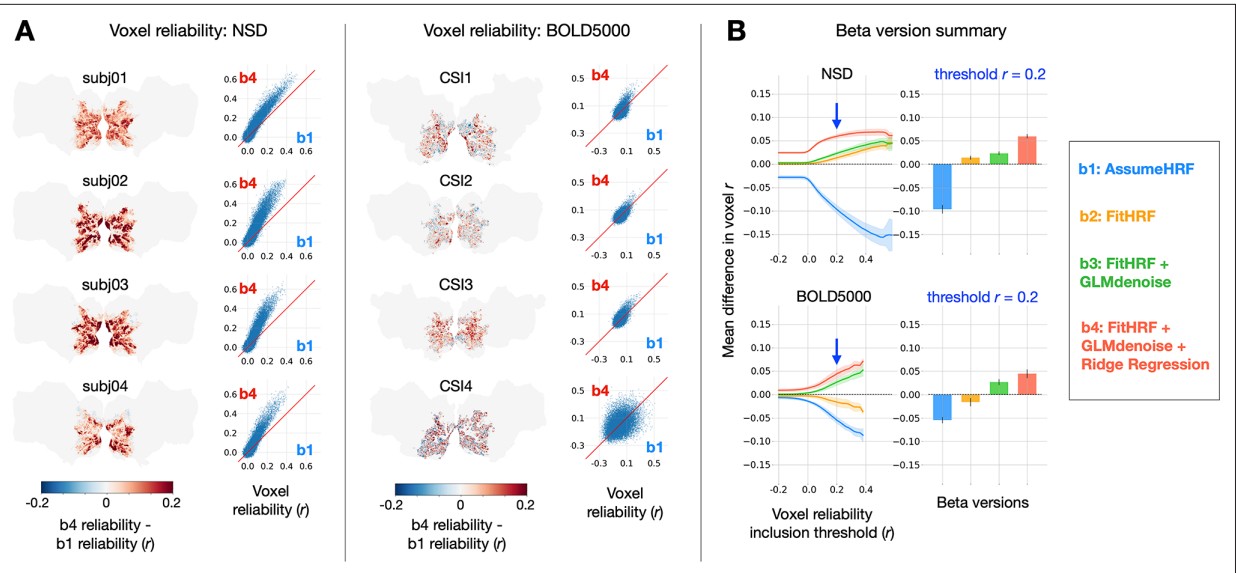

**Figure 2.** Impact of GLMsingle on voxel test-retest reliability. To compute reliability for a given voxel, we measure the test-retest Pearson correlation of GLM beta profiles over repeated presentations of the same stimuli (see Materials and methods). (**A**) Differences in reliability between *b1* (derived from a baseline GLM) and *b4* (the final output of GLMsingle) are plotted within a liberal mask of visual cortex (nsdgeneral ROI). Scatter plots show reliability values for individual voxels. (**B**) Relative differences in mean reliability within the nsdgeneral ROI. For each voxel, we computed the mean reliability value over all beta versions being considered (*b1-b4*), and then used this as the basis for thresholding voxels (from Pearson $r = –0.2 – 0.6$). At each threshold level, for each beta version, we compute the voxel-wise difference between the reliability of that specific beta version and the mean reliability value, and then average these difference values across voxels within the nsdgeneral ROI. The traces in the first column indicate the mean (+/- SEM) across subjects within each dataset (N = 4 for both NSD and BOLD5000). The bars in the second column indicate subject-averaged differences in reliability at threshold $r = 0.2$. The relative improvement in reliability due to GLMsingle (*b1* vs. *b4*) tends to increase when examining voxels with higher reliability, and each optimization stage within GLMsingle (HRF fitting, GLMdenoise, ridge regression) confers added benefit to voxel reliability.

The online version of this article includes the following figure supplement(s) for figure 2:

**Figure supplement 1.** Inspection of HRF structure across space and time. Here we examine the optimal HRF indices chosen by GLMsingle within a liberal mask of visual cortex (nsdgeneral ROI) from an example subject (NSD subj01).

HRF that provides the best fit to the data (highest variance explained), and inherit the single-trial betas associated with that HRF (*b2*: FitHRF).

3. GLMdenoise (***Kay et al., 2013***; ***Charest et al., 2018***) is used to determine nuisance regressors to include in the model. Principal components analysis is applied to time-series data from a pool of noise voxels (see Materials and methods for details), and the top principal components are added one at a time to the GLM until cross-validated variance explained is maximized on-average across voxels (*b3*: FitHRF + GLMdenoise).

4. With the nuisance regressors determined, fractional ridge regression (***Rokem and Kay, 2020***) is used to regularize the single-trial betas, using a custom amount of regularization for each voxel, determined via cross-validation (*b4*: FitHRF + GLMdenoise + RR).

## GLMsingle improves the reliability of beta estimates

We first examined the effect of GLMsingle on the test-retest reliability of voxels across relevant regions of visual cortex in NSD and BOLD5000 (***Figure 2***). For NSD, we analyzed the first 10 scan sessions of data from each of the first 4 subjects (each scan session consisted of 12 x 5.0 min runs). Each scan session was analyzed separately, and each scan session contained, on average, 35 conditions with 3 trials each, 107 conditions with 2 trials each, and 431 conditions with 1 trial each. For BOLD5000, we analyzed the complete dataset, consisting of subjects CSI1-4. These data were collected over 15 scan sessions (each scan session consisted of either 9 or 10 runs lasting 6 min and 28 s each). Groups of 5 scan sessions were analyzed jointly, and each group resulted in, on average, 0 conditions with 4 trials each, 9 conditions with 3 trials each, 40 conditions with 2 trials each, and 1642 conditions with 1 trial each. Subject CSI4 completed only 9 of 15 experimental sessions, and their data were analyzed in two partitions of 5 and 4 sessions each. Our reliability procedure measures the consistency of a voxel's

response profile (using Pearson *r*) over repeated presentations of the same stimuli, revealing areas of the brain containing stable BOLD responses. This straightforward approach enables direct comparison of data quality between different beta versions.

We directly compared the *b*1 and *b*4 beta versions for each subject within a liberal mask of visual cortex (nsdgeneral ROI), finding widespread increases in reliability when comparing GLMsingle to baseline (*Figure 2a*). The positive effect is nearly uniform across voxels in NSD. In BOLD5000, as in NSD, we see aggregate benefits when comparing *b*1 and *b*4, though results for individual voxels are more variable. A likely explanation for this is that reliability metrics are inherently noisier due to the smaller number of repeated stimuli in BOLD5000. In addition, the magnitude of the benefits of *b*4 over *b*1 are somewhat smaller in BOLD5000 compared to NSD. This is likely true for a number of reasons, including the generally lower SNR in BOLD5000 (as a data-driven technique, GLMsingle is more effective at higher levels of SNR) and the relatively long inter-stimulus interval in BOLD5000 (ridge regression is expected to have smaller impact when there is less overlap of the BOLD response across successive trials).

To summarize the impact of GLMsingle in NSD and BOLD5000, we compared the performance of *b*1-*b*4 for individual subjects, across different voxel reliability thresholds (*Figure 2b*). We find that all subjects show clear improvement from *b*1 to *b*4 and that the improvement in reliability due to GLMsingle tends to increase when examining voxels that respond more reliably to experimental stimuli. Furthermore, examining reliability in intermediate beta versions (*b*2 and *b*3) – which implement HRF optimization and GLMdenoise, respectively – reveals that each successive stage of processing in GLMsingle tends to confer added benefit to voxel reliability compared to baseline (*b*1).

To better understand the nature of the HRF fitting procedure, we examined selected HRFs across visually-responsive cortex of a representative subject (*Figure 2—figure supplement 1*). In active voxels, we observe a structured, low-frequency spatial gradient in HRF indices (*Figure 2—figure supplement 1b*). Moreover, identified HRF indices are highly consistent from session to session (*Figure 2—figure supplement 1c*). This provides evidence that, beyond merely capturing subject-specific deviation from the assumed HRF, the FitHRF procedure confers benefit by capturing voxel-wise differences in HRF shapes. We note that there are strong biophysical reasons to expect voxel-wise differences in HRF shapes, related to variation in the brain's vasculature (*Kay et al., 2020*).

We next compared GLMsingle to Least-Squares Separate (LSS), a popular technique for robust signal estimation in rapid event-related designs (*Mumford et al., 2012*; *Mumford et al., 2014*; *Abdulrahman and Henson, 2016*). The LSS procedure fits a separate GLM for each stimulus, where the trial of interest is modeled as one regressor, and all other (non-target) trials are collapsed into a second regressor. LSS provides a useful point of comparison for ridge regression, as both strategies seek to mitigate the instabilities in GLM estimation that can arise from having correlated single-trial predictors. To directly compare GLMsingle to LSS, we computed auxiliary GLMsingle beta versions that do not incorporate GLMdenoise. This allows us to isolate the effect of the GLM estimation procedure (i.e. LSS vs. fractional ridge regression).

For both the case of an assumed HRF and the case of voxel-wise tailored HRFs, we find that fractional ridge regression yields more reliable signal estimates than LSS (*Figure 3*). These improvements are most pronounced in the most reliable voxels (*Figure 3c*). LSS can be viewed as applying heavy regularization uniformly across voxels, while our ridge regression approach is more flexible, tailoring the degree of regularization to the SNR of each voxel. Heavy regularization may actually degrade the quality of signal estimates in reliable voxels, and our approach avoids this possibility.

We then performed a complete assessment of all auxiliary beta versions and the primary versions (*b*1-*b*4), in order to determine whether any other analysis strategy could achieve parity with *b*4 in the quality of GLM outputs. Reassuringly, when summarizing the relative quality of all 8 beta versions over a range of reliability thresholds, we observe superior performance from *b*4, the default output of GLMsingle (*Figure 3a*). For these and subsequent findings, it is important to note that differences between beta versions should be interpreted with caution at very high reliability thresholds, as voxel counts may be quite low for certain subjects.

GLMsingle relies on an internal cross-validation procedure through which key hyperparameters (the number of noise regressors and the voxel-wise levels of ridge regression regularization) are optimized to maximize the consistency of responses across condition repetitions. This raises a possible concern that our reliability estimates (e.g. *Figure 2*) are somewhat optimistic. As a strict assessment of

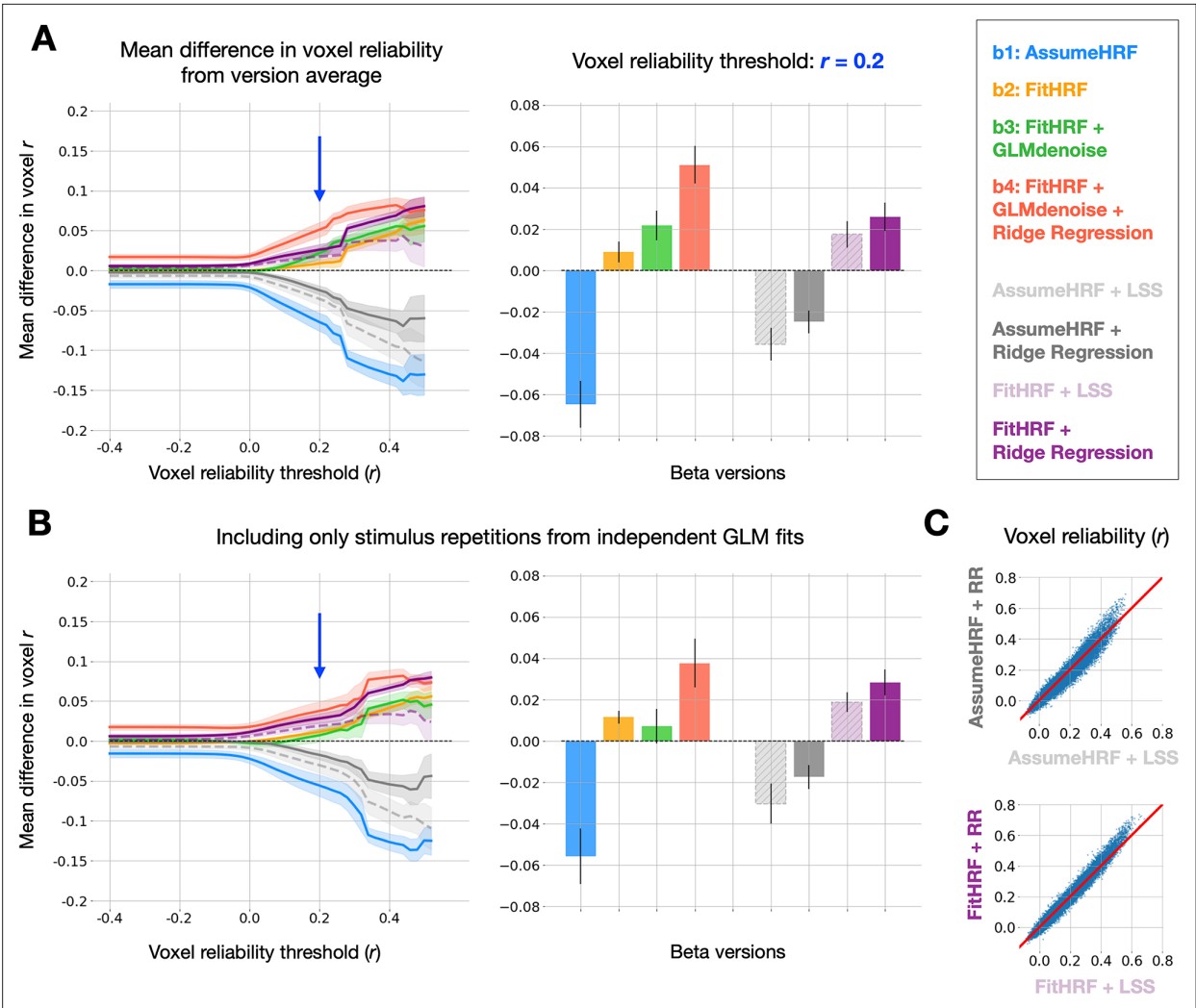

**Figure 3.** Relative quality of GLMsingle and LSS beta versions. (**A**) Left panel: relative differences in mean reliability between beta versions. 8 beta versions are compared: $b1$-$b4$, and the 4 auxiliary beta versions used to compare GLMsingle and Least-Squares Separate (LSS). LSS betas (dashed traces) are compared to those estimated using fractional ridge regression (RR, solid traces), when using a canonical HRF (LSS, light gray vs. RR, dark gray) and when performing HRF optimization (LSS, light purple vs. RR, dark purple). Right panel: summary of performance at threshold level $r = 0.2$. Error bars reflect the standard error of the mean, computed over the 8 subjects analyzed from NSD and BOLD5000. Fractional ridge regression yields more reliable signal estimates than LSS across voxel reliability levels. (**B**) Same as Panel (**A**), except that reliability computations occur only between image repetitions processed in independent partitions of fMRI data. Qualitative patterns are unchanged. (**C**) Scatter plots comparing voxel reliability between corresponding LSS and GLMsingle beta versions (top: AssumeHRF; bottom: FitHRF). Plotted are results for an example subject (NSD subj01, nsdgeneral ROI). The advantage of ridge regression over LSS is most apparent in the most reliable voxels.

reliability, we repeated the reliability quantification for each of the 8 beta versions, this time computing test-retest correlation values using only beta responses obtained from completely separate data partitions. We find that results are broadly unchanged using this more stringent evaluation procedure (*Figure 3b*).

As a further test of the general applicability of GLMsingle, we repeated the above procedures using data from the music-listening component of StudyForrest (*Hanke et al., 2015*). This dataset measures brain responses as subjects listen to 25 distinct 6-s musical clips from 5 genres, with 8 trials per condition for each of 16 subjects. Each condition occurred once per functional run, and each subject completed one session of data consisting of 8 runs. This dataset differs from NSD and BOLD5000 in several key respects: the scale (there are far fewer trials), the task modality (auditory, as opposed to visual), and the use of a jittered inter-stimulus interval (the delay between trials is variable between 4 and 8 s). As in NSD and BOLD5000, we observe substantial improvements in

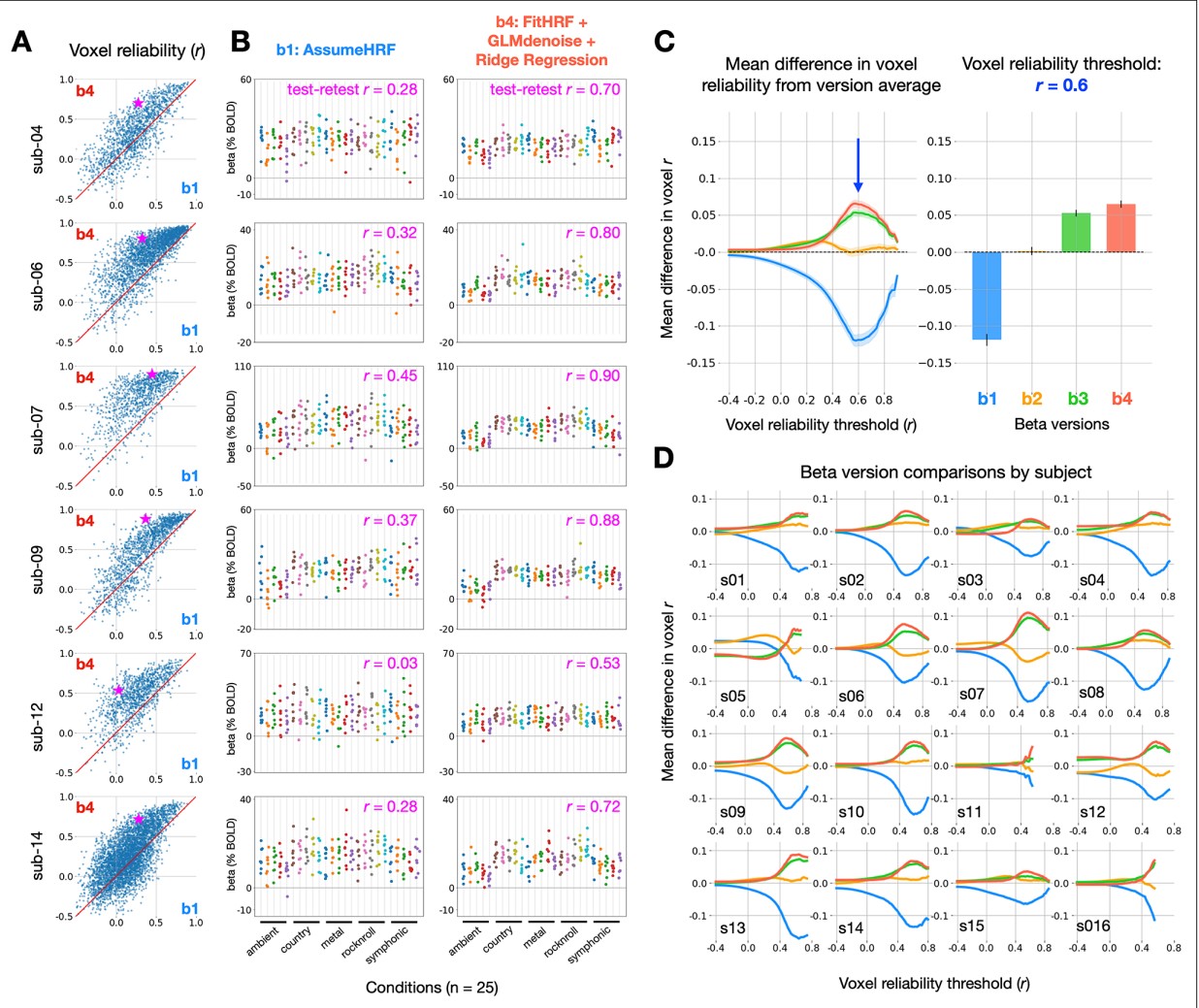

**Figure 4.** Impact of GLMsingle on reliability in the *StudyForrest* music-listening task. (**A**) Differences in voxel test-retest reliability (Pearson *r*) between *b*1 (a baseline GLM) and *b*4 (the final output of GLMsingle) are plotted for individual voxels. Only voxels that are active in response to experimental stimuli (ON-OFF $R^2 > 5$) are plotted. (**B**) Estimated beta values (% BOLD change) for *b*1 and *b*4 in a hand-selected auditory cortex voxel from 6 representative subjects. Chosen voxels are indicated with pink stars in panel A. Each column represents one of 25 experimental conditions, with each condition presented 8 times. Test-retest reliability values reflect the split-half correlation between groups of 4 trial repetitions, averaged over all possible splits of the available repetitions (70 unique splits). (**C**) Relative differences in mean reliability between beta versions *b*1 - *b*4, computed using the same procedure as used for NSD and BOLD5000 (see *Figure 2*). Traces indicate the mean (+/- SEM) across subjects (N = 16). The bar graph (right) indicates the subject-averaged differences in reliability at threshold $r = 0.6$. (**D**) Relative differences in mean reliability over different reliability inclusion thresholds are plotted for each subject.

reliability through the application of GLMsingle (*Figure 4a–c*), these improvements are consistent across subjects (*Figure 4d*), and each individual component of GLMsingle confers added benefit in reliability compared to the baseline GLM. These findings, arising from a smaller scale dataset that may more closely resemble a typical fMRI study, suggest the general applicability of GLMsingle to a wide range of datasets.

## GLMsingle helps disentangle neural responses to neighboring trials

Thus far, we have established that GLMsingle provides BOLD response estimates that have substantially improved reliability compared to a baseline GLM. In the remainder of this paper, we explore whether these improvements have tangible consequences for downstream analyses relevant to cognitive and systems neuroscience. We first examine whether GLMsingle is able to more effectively disentangle neural responses to proximal stimuli, as inaccurate single-trial GLM estimation may manifest

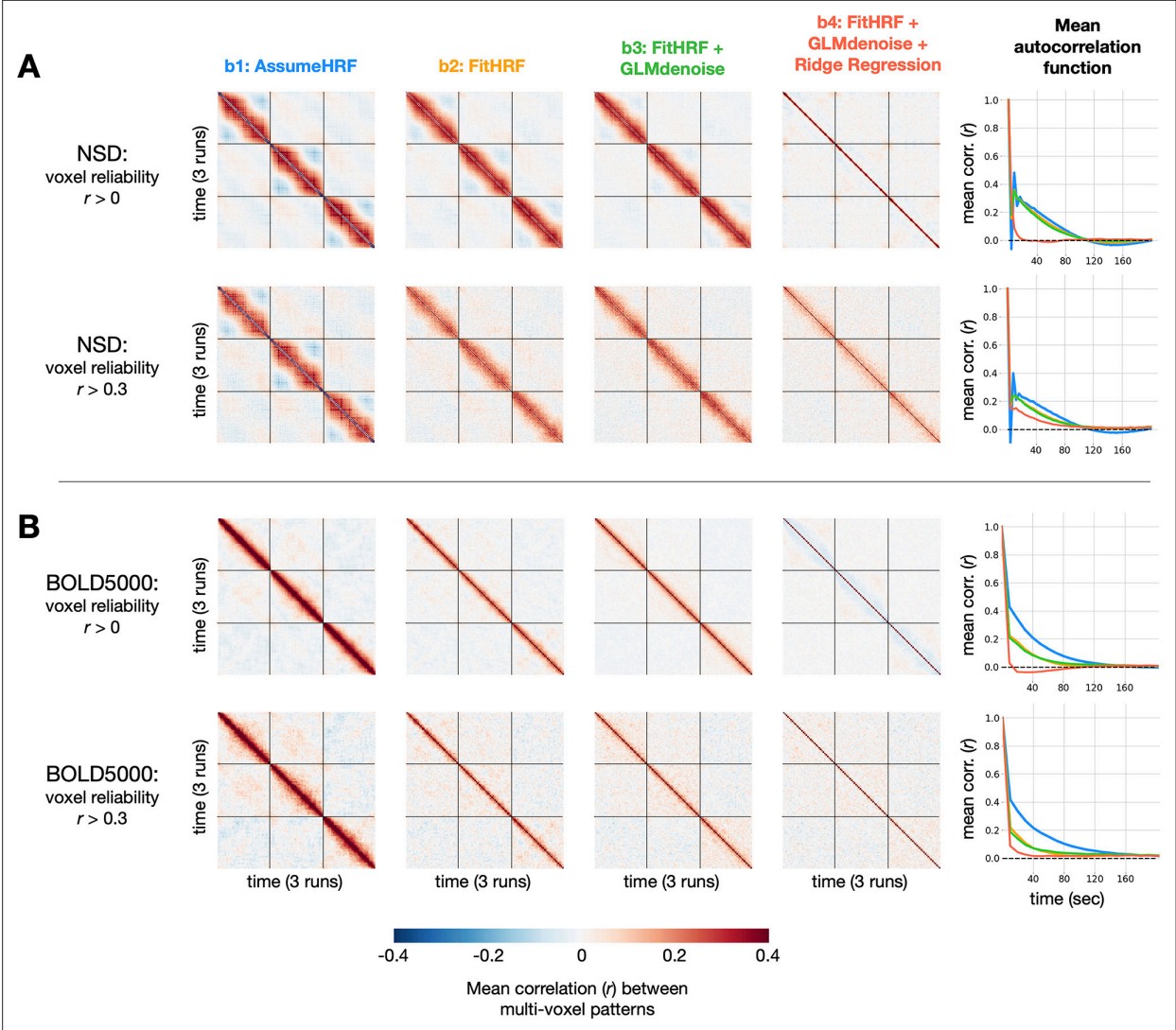

**Figure 5.** Impact of GLMsingle on temporal autocorrelation. For each dataset, we compute the degree of temporal autocorrelation in each beta version by averaging session-wise representational similarity matrices over subjects. We plot results arising from analysis of voxels at two different reliability thresholds ($r = 0$ and $r = 0.3$) for NSD (**A**) and BOLD5000 (**B**). Assuming that ground-truth neural responses to consecutive trials should be uncorrelated on average, positive (or negative) Pearson $r$ values off the diagonal imply suboptimal estimation of BOLD responses. In the right-most column, we plot mean autocorrelation between all pairs of timepoints. Applying GLMsingle ($b4$) results in a substantial decrease in temporal autocorrelation compared to a baseline GLM approach ($b1$).

as high similarity (temporal autocorrelation) between beta maps from nearby trials. We computed dataset-averaged temporal similarity matrices, revealing the degree of temporal autocorrelation in each beta version (*Figure 5*). Temporal autocorrelation manifests as non-zero correlation values off the diagonal of the temporal similarity matrices, and is presumably undesirable.

In a baseline GLM that uses a canonical HRF and ordinary least-squares (OLS) fitting ($b1$), we observe striking patterns of temporal autocorrelation extending several dozen trials forward in time. This is true in both NSD, which has a rapid event-related design (a new stimulus presented every 4 s), as well as in BOLD5000, where stimuli are spaced 10 s apart to alleviate issues relating to signal overlap. To quantify these effects, we compute mean temporal autocorrelation as a function of time post-stimulus for each beta version. In NSD, for the baseline GLM ($b1$), positive correlations are as high as $r = 0.5$ for consecutive trials, and gradually reduce to around $r = 0$ after around 100 s (*Figure 5a*). In BOLD5000, $b1$ autocorrelation peaks as high as around $r = 0.4$ for consecutive trials, requiring nearly 160 s to reduce to $r = 0$ (*Figure 5b*). We speculate that the relatively long timescale of the correlations reflects the long timescale of hemodynamic responses (the post-undershoot can extend

for 30 s or longer) and/or the slow nature of (low-frequency) physiological noise related to cardiac and respiratory variation. Notably, mean beta maps from successive trials in NSD are *anticorrelated* for $b1$, a known artifact of OLS fitting in the case of high multicollinearity between GLM predictors (*Mumford et al., 2014*; *Soch et al., 2020*).

When applying GLMsingle, these patterns of temporal autocorrelation change dramatically. In NSD $b4$, autocorrelation drops to $r = 0$ much more rapidly than in $b1$, and in BOLD5000, beta maps from successive trials in $b4$ are now nearly uncorrelated on average. This is an expected outcome, since the stimuli in NSD and BOLD5000 are ordered pseudorandomly. In both datasets, an intermediate beta version ($b2$) containing only HRF optimization confers marginal benefit over $b1$, but the most dramatic improvements come from the addition of both GLMdenoise and fractional ridge regression ($b4$). Overall, these results demonstrate the utility of GLMsingle for disentangling neural responses to nearby stimuli in event-related designs, even when events are presented relatively slowly (as in BOLD5000).

## GLMsingle improves between-subject representational similarity across datasets

Large-scale datasets such as NSD and BOLD5000 are well-suited for representational analyses (e.g. RSA) that compare evoked neural response patterns between individual subjects, across different experimental modalities, and against computational models (e.g. deep neural networks, see

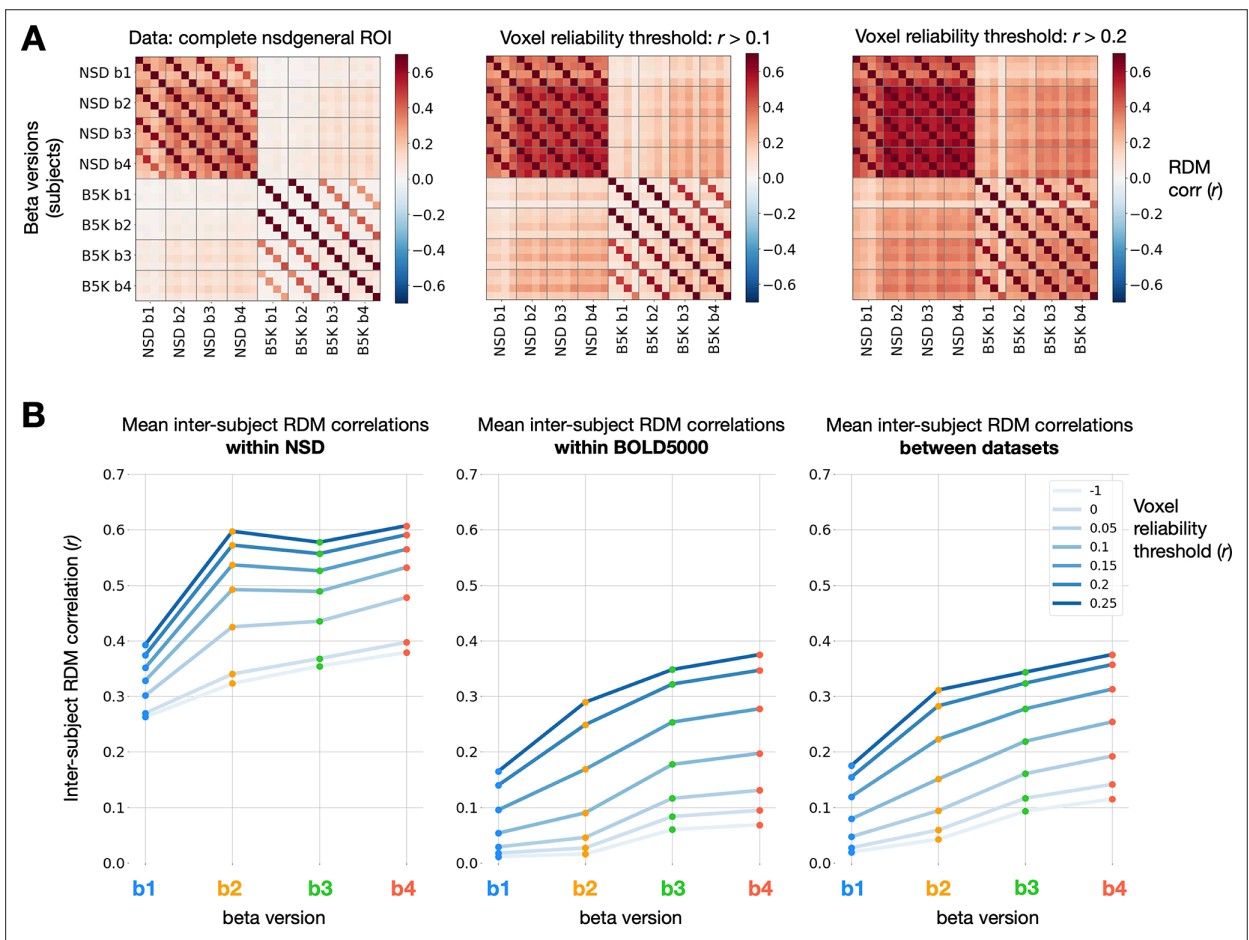

**Figure 6.** Impact of GLMsingle on inter-subject RDM correlations. (**A**) Correlations of RDMs across all pairs of subjects and beta versions, at 3 different voxel reliability thresholds. We compute RDMs for each subject and beta version using Pearson dissimilarity (1 - *r*) over repetition-averaged betas within the nsdgeneral ROI. Grid lines separate beta versions from one another, an individual cell reflects the RDM correlation between one pair of subjects, and cross-dataset comparisons occupy the top-right and bottom-left quadrants of the matrices. (**B**) Mean inter-subject RDMs correlations within NSD (N = 4; left), within BOLD5000 (N = 4; center), and between the two datasets (N = 16 subject pairs; right). GLMsingle (*b4*) yields a considerable strengthening of RDM correspondence for each subject pair being considered, within and between datasets.

*Kriegeskorte, 2015*; *Serre, 2019* for review.) In almost all such studies, representational analyses presume that the same set of stimuli will evoke reasonably similar responses across subjects. As such, given the ubiquity of noise in fMRI, it is reasonable to expect that improving the accuracy of single-trial response estimates should yield representations that are more similar across individuals.

To compare representations between subjects, we used the approach of RSA (*Kriegeskorte et al., 2008*). First, we isolated stimuli that overlap between BOLD5000 and the subset of NSD analyzed for this manuscript (the first 10 sessions from each subject). Using these 241 stimuli, we constructed representational dissimilarity matrices (RDMs) using repetition-averaged betas from each individual, and then correlated all pairs of subject RDMs within and between datasets. Note that GLMsingle is not designed to enhance or optimize cross-subject representational similarity; as such, it is informative to examine RSA correlations between subjects as a way of assessing methods for denoising (*Charest et al., 2018*). Strikingly, in comparing beta versions $b1$ and $b4$, we observe a consistent strengthening of RDM correspondence (*Figure 6b*). This trend held within NSD, within BOLD5000, and when comparing the RDMs of subject pairs between the two datasets. The latter result is especially notable given the many methodological differences between NSD and BOLD5000: fMRI data were collected at different sites on different scanners, at different field strengths (7T vs. 3T), with different behavioral tasks, and with different inter-stimulus intervals (4 s vs. 10 s).

These results indicate that GLMsingle, through its multifaceted approach to mitigating the effects of noise, helps reveal meaningful shared variance in neural responses across individuals who viewed the same stimuli. The GLMsingle toolbox may therefore be a key resource for future fMRI studies seeking to stitch together data across subjects from different sites or cohorts.

## GLMsingle enables fine-grained image-level MVPA decoding

As a final analysis, we assessed the effect of GLMsingle on the results of multivoxel pattern analysis (MVPA). In a "one-vs.-many" classification paradigm, we trained linear SVM models for each subject to predict image identity from neural response patterns. The baseline GLM ($b1$) classification accuracy was slightly above chance on average for the subjects in NSD and BOLD5000 when including all visual cortex voxels (*Figure 7a*, blue traces). Performing the same MVPA procedure using GLMsingle betas ($b4$), we observe that mean accuracy approximately triples in NSD and doubles in BOLD5000 (*Figure 7a*, red traces). Moreover, in both datasets, we observe a substantial increase in classification accuracies with increasing voxel reliability threshold, with the most dramatic improvements achieved using $b4$ in NSD (*Figure 7a*, left panel, right-most bins).

The level of performance that GLMsingle facilitates on this challenging multi-way decoding task highlights the ability of the technique to accurately identify and model the stable structure contained in noisy fMRI time-series. To illustrate this point, we performed 2D multidimensional scaling (MDS, *Borg and Groenen, 2005*) using the NSD betas that were included in MVPA. Comparing results between beta versions $b1$ and $b4$, we observe improved clarity of an animacy division in the representational space of an example subject (*Figure 7b*).

## Discussion

As scientific datasets grow in scale and scope, new techniques for data processing will help to unlock their potential. This is especially true in human neuroscience where data remain both expensive and time-consuming to collect (*Naselaris et al., 2021*). This paper has introduced GLMsingle, a publicly available toolbox for analyzing fMRI time-series data that leverages data-driven techniques to improve the accuracy of single-trial fMRI response estimates. We have tested GLMsingle extensively using NSD and BOLD5000, two of the largest fMRI datasets that densely sample responses within individuals. For both datasets, analyses of the response estimates provided by GLMsingle indicate substantial improvements in several key metrics of interest to neuroscientists: (i) enhanced test-retest reliability of voxel response profiles, a straightforward metric of data quality; (ii) reduced temporal autocorrelation, a common fMRI effect that is presumably undesirable and especially prominent in rapid event-related designs; (iii) increased representational similarity across subjects both within and across datasets; and (iv) improved multivariate pattern classification performance when discriminating responses evoked by individual images.

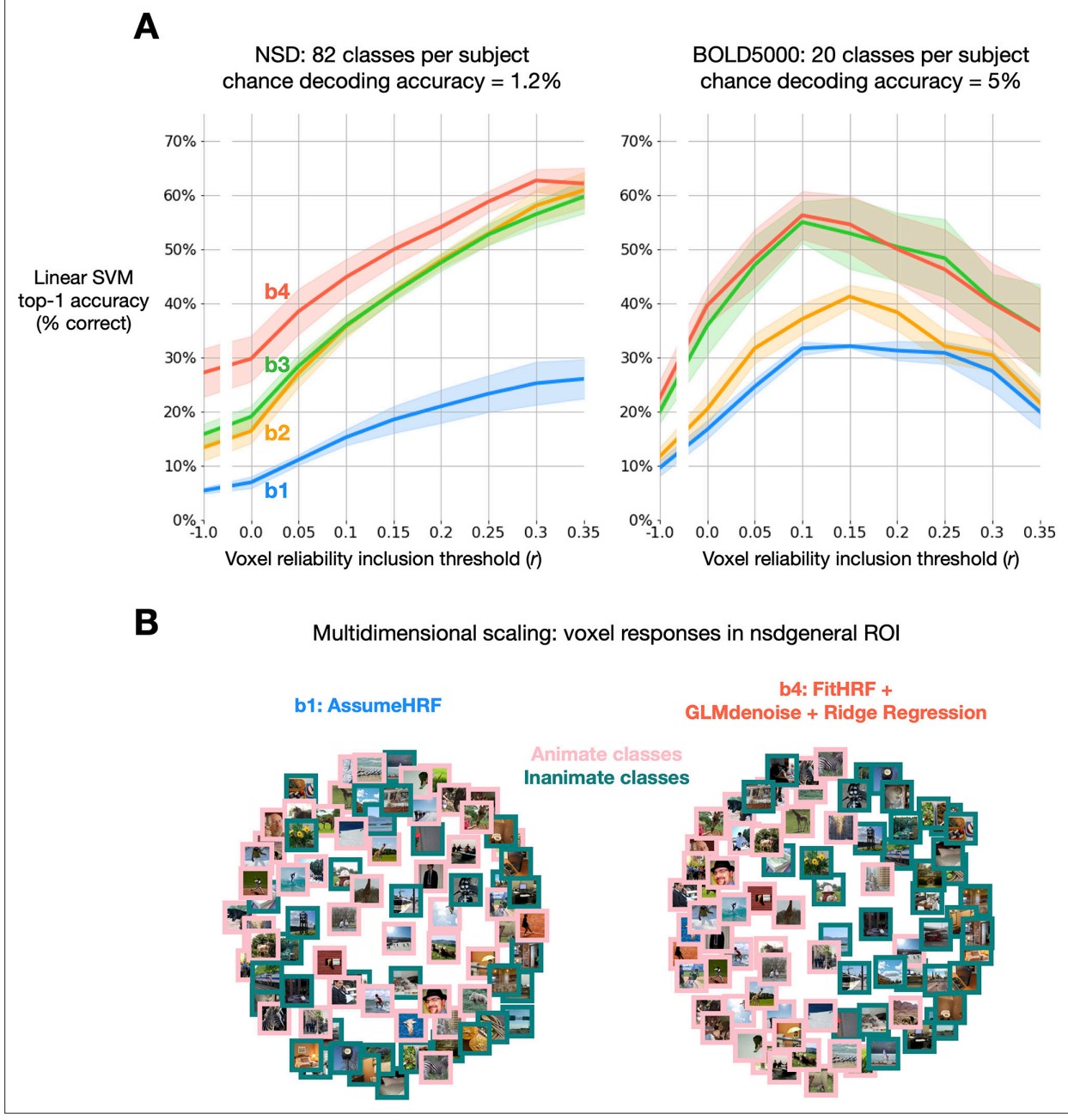

**Figure 7.** Impact of GLMsingle on image-level MVPA decoding accuracy. (**A**) Image-level linear SVM decoding accuracy by beta version. At each reliability threshold, we compute the mean decoding accuracy over subjects within each dataset, as well as the standard error of the mean (N = 4 for NSD; N = 3 for BOLD5000). Classifiers are trained on $n - 1$ available image repetitions, and tested on the held-out repetition, with accuracy averaged over cross-validation folds. Applying GLMsingle (*b4*) yields dramatic increases in image decodability compared to a baseline GLM (*b1*). (**B**) The effect of GLMsingle on animacy representation is shown in an example NSD subject (subj01) using multidimensional scaling. GLMsingle clarifies the division in representational space between stimuli containing animate and inanimate objects.

## Principles underlying GLMsingle

GLMsingle incorporates three optimization procedures to improve the estimation of fMRI responses:

1. *HRF fitting.* GLMsingle uses a "library of HRFs" technique to select the most appropriate HRF to use for each voxel in a given dataset (*Allen et al., 2022*). This library consists of a set of 20 HRFs that were derived from experimental data (specifically, the first NSD scan session acquired in each of the eight NSD subjects). It is well known that variations in HRFs exist across voxels, brain areas, and subjects, and that mismodeling the timecourse of a voxel may lead to suboptimal

analysis outcomes (*Handwerker et al., 2004*; *Handwerker et al., 2012*). Imposing constraints on HRF selection by choosing from a fixed set of HRFs avoids the instability (high variance) associated with more flexible timecourse modeling approaches, such as finite impulse response modeling (*Kay et al., 2008*; *Bai and Kantor, 2007*). Variations in timecourse shapes in the HRF library reflect a continuum between short-delay, narrow-width timecourses to long-delay, broad-width timecourses, and are likely caused by variations in the contribution of large vessels to the BOLD response observed in a voxel (*Kay et al., 2020*).

2. *Data-driven denoising.* Incorporating an adaptation of the GLMdenoise technique (*Kay et al., 2013*), GLMsingle uses principal components analysis to calculate potential nuisance regressors from fMRI time-series data observed in voxels that are deemed unrelated to the experimental paradigm. These regressors are incorporated into the GLM using a cross-validation procedure to determine the optimal number of nuisance regressors to add. A key aspect of our approach is the acknowledgement that including increasing numbers of nuisance regressors will, at some point, cause overfitting and degradation of results (*Kay et al., 2013*); this motivates the use of cross-validation to determine the optimal level of model complexity.

3. *Regularization of GLM weights.* To improve the accuracy of single-trial GLM response estimates, GLMsingle uses fractional ridge regression (*Rokem and Kay, 2020*), with an optimal degree of regularization identified for each voxel, again using cross-validation. The improvements afforded by this procedure are due to the substantial amount of overlap of the fMRI response across successive trials, unless very long (>30 s) inter-stimulus intervals are used. It is well known that, in the context of ordinary least-squares estimation, two predictors that are correlated (or anti-correlated) will have reduced estimation precision compared to the scenario in which the predictors are uncorrelated (*Mumford et al., 2012*; *Soch et al., 2020*). For rapid event-related designs, predictors for consecutive trials are typically correlated, and ordinary least-squares estimates will suffer from high levels of instability. Ridge regression imposes a shrinkage prior (penalizing the sum of the squares of the beta estimates), which can, in principle, dampen the effects of noise and improve out-of-sample generalizability of the beta estimates.

## Ideal use-cases for GLMsingle

GLMsingle is designed to be general in its application. It uses data-driven procedures that automatically adapt to the signal-to-noise characteristics of a given dataset. For example, in datasets where structured noise is prevalent, appropriate nuisance regressors will automatically be included, whereas in datasets with very little structured noise (e.g. low head motion), fewer (or no) nuisance regressors will be included. As another example, for experimental designs with high temporal overlap between consecutive trials or high levels of noise, relatively strong levels of shrinkage regularization will likely be selected.

GLMsingle is a general technique that can be fruitfully applied to nearly *any* fMRI experiment involving discrete events (including block designs). However, we recognize that integrating a new tool into an analysis workflow requires effort. Therefore, we anticipate that the most consequential impact of GLMsingle will be observed for study designs with low sensitivity (such as condition-rich designs).

## Implementation guidelines

Datasets can have complex features that may complicate the way in which one applies GLMsingle. In this section, we comment on major implementation-related choices that the user may face when deciding how to apply GLMsingle to their own experimental paradigms. Additional discussion of more minor issues can be found in the online toolbox documentation.

### Experimental design
#### Requirement of repeated conditions

In order to determine appropriate hyperparameter settings, it is necessary for some number of repeated trials to exist in the data that are input to GLMsingle. It is not critical that the conditions be balanced in their number of repetitions, and even a handful of repeats appears to be sufficient to robustly guide the estimation of hyperparameters. For example, for the groups of sessions from BOLD5000 that were analyzed in this paper, typically only about 3% of the conditions were repeated (i.e. had more than a single trial).

## Minimum trial separation

The degree to which GLMsingle can effectively separate response amplitudes to closely spaced trials is an empirical question that depends, in part, on the temporal characteristics of the hemodynamic response function. As shown in this paper, effective results were obtained in the Natural Scenes Dataset where the trial separation is 4 s (images are shown for 3 s and the inter-stimulus interval is 1 s). More rapid experimental designs will generally incur greater overlap in the BOLD responses to neighboring trials, making response estimation more challenging. However, in these scenarios a greater number of trials can be run, which may counteract the loss of power arising from the response overlap. Future work could seek to systematically manipulate inter-trial intervals to assess the effectiveness of GLMsingle under more rapid designs.

## Coding of blank trials

GLMsingle expects that blank trials are not coded in the design matrix. The GLM uses a set of polynomials per run to model the baseline signal, which is associated with sections of the experiment that are not explicitly coded as events. Single-trial betas therefore reflect evoked BOLD responses above and beyond the baseline signal. Coding blank trials as events would interfere with this important component of the GLM estimation procedure.

## Modeling trial subcomponents

In some experiments, there may be multiple stages of a given trial (e.g. cue, preparatory period, stimulus presentation). GLMsingle does not distinguish between these subcomponents, and the particular modeling strategy is at the discretion of the experimenter. One approach is to treat all subcomponents of a trial as contributing to a single response amplitude. Alternatively, each subcomponent can be treated as a distinct condition (e.g. the cue for condition A coded as "condition A1" and the stimulus for condition A coded as "condition A2", and so on). In such an approach, the goal is to estimate a separate response amplitude for each subcomponent of each trial.

## fMRI data preparation

### Handling data from multiple scan sessions

BOLD responses and noise characteristics can change substantially across scan sessions (days) for an individual. For this reason, it is generally recommended to apply GLMsingle to individual scan sessions, if possible. However, if it is necessary to apply GLMsingle to data concatenated across days, one potential problem is that gross amplitude differences across sessions will be treated as noise, leading to suboptimal estimation performance. Users may avoid this issue by taking advantage of the *sessionindicator* option, which allows users to specify how different input runs are grouped into sessions. GLMsingle uses this information internally to $z$-score responses within sessions to better estimate cross-validated model performance. This normalization step does not propagate to final outputs. In the processing of BOLD5000 data for this manuscript, we indeed took advantage of the *sessionindicator* option (see Materials and methods).

### Compatible pre-processing steps

Beyond the expectation that the fMRI data provided to GLMsingle are minimally pre-processed (e.g. motion corrected and ideally slice-time corrected), there are several other common pre-processing steps that are fully compatible with GLMsingle. First, the pipeline works in a straightforward manner with data that have been masked, skull-stripped, or projected into surface space (where voxels falling outside of gray matter may have already been excluded). Second, spatial changes (such as the registration of fMRI data to an atlas space, or to a subject's anatomy) do not interfere with the operation of GLMsingle. Third, it is acceptable to perform grand intensity scaling to normalize units across subjects and/or sessions. However, users should be wary of scaling procedures that are applied differently across brain regions or voxels, or across different runs in a session. Finally, since GLMsingle requires the design matrix and fMRI data to be temporally synchronized, users may need to resample one or both objects in order to match their temporal resolution.

## Incompatible pre-processing steps

Several common fMRI pre-processing steps are unnecessary in conjunction with GLMsingle. For example, high-pass filtering of fMRI data is not recommended, as GLMsingle automatically includes a set of polynomials to model low-frequency drifts in the baseline signal that may occur over the course of each run. Neither is low-pass filtering necessary, since the use of well-regularized hemodynamic response functions to model evoked responses automatically discounts high-frequency temporal noise. In addition, it is not recommended to project out putative nuisance components of fMRI data (e.g. motion regressors, nuisance regressors derived from white matter or cerebrospinal fluid) prior to running GLMsingle, as these pre-filtering approaches pose a risk of biasing signal estimates. GLMsingle's approach is to learn nuisance components from the data and to respect the potential overlap of these components with signal components of interest.

## HRF estimation
### HRF selection does not require cross-validation

The components of GLMsingle beyond HRF selection each have natural, unregularized states: for GLMdenoise, this entails including no additional nuisance regressors, and for ridge regression, this entails performing ordinary least-squares estimation and adding no shrinkage bias. The selection of the HRF, however, is different. At least in its current formulation, GLMsingle views each of the HRFs from the library as an equally valid HRF, which is justifiable given the diversity of HRF shapes across voxels and subjects. Thus, there is no notion of regularization inherent to the procedure, and GLMsingle needs only to select the HRF that maximizes fit to the data.

## GLMdenoise
### Control over the noise pool

The default behavior of GLMsingle is to automatically select noise pool voxels as those that both exceed a simple signal intensity threshold (which excludes out-of-brain voxels) and have negligible amounts of BOLD variance related to the experiment (using an $R^2$ threshold). If desired, users of GLMsingle can control the noise pool voxels via the *brainthresh* argument (determining which voxels pass the intensity threshold), the *brainR2* argument (determining which voxels pass the $R^2$ threshold), and the *brainexclude* argument (allowing for a custom mask of voxels to be considered for the noise pool). Users may also limit the number of noise PCs that may be added to the GLM (or hard-code the quantity ahead of time) via the *pcstop* argument.

### Constraints on denoising

Could denoising remove signals of interest? GLMsingle guards against improper use of nuisance regressors through cross-validation. If, for any reason, there are valid experimental signals being derived from the noise pool, GLMsingle will tend to avoid including these regressors since they will likely degrade the cross-validation performance of the model. In fact, even if the noise pool is deliberately expanded to include signal-carrying voxels, some improvement in beta estimates is still possible, since the GLM estimation can theoretically separate the variance contributions from signal and nuisance regressors (*Kay et al., 2013*).

## Fractional ridge regression
### Scale and offset following regularization

By default, after the application of ridge regression, GLMsingle applies a post-hoc scale and offset to the single-trial betas obtained for a given voxel to best match the response distribution of the unregularized case. The reason for this is that due to shrinkage, the betas for voxels with poor signals are shrunken close to 0 – they have a bias to be small in magnitude. The simple scale and offset step is intended to undo this bias. Users may omit the scale and offset (via appropriate setting of the *wantautoscale* argument) and/or avoid ridge regression altogether (via *wantfracridge*).

### Scenarios where key hypotheses depend on neighboring trials

Studies seeking to analyze responses to trials that are close together in time (e.g. repetition suppression tasks, or preparation-execution in motor tasks) pose fundamental challenges with respect to

signal estimation, due to the substantial overlap in BOLD responses when events occur within seconds of one another. Users should be mindful of how they analyze data using GLMsingle in these contexts. Of particular relevance is the fact that ridge regression, through its shrinkage bias, encourages some amount of temporal smoothing of beta estimates for nearby trials (in the sense that beta weights from temporally adjacent trials are biased to be more similar in magnitude). When the critical hypotheses depend on responses to nearby trials, users may consider disabling the ridge regression component of GLMsingle (to avoid any effects of temporal smoothing). However, note that there is no guarantee that doing so will improve the accuracy of signal estimation – single-trial predictors associated with neighboring events will still be highly correlated, and ordinary least-squares (OLS) estimates will exhibit structured, correlated errors as a result. In fact, our empirical results suggest that ridge regression reduces unwanted temporal autocorrelation (*Figure 5*).

## Ridge regression enables interpretation of GLM $R^2$

In the case of a single-trial design matrix, the quantity and flexibility of the predictors enables a regression model fit using OLS to capture almost all the variance in the time-series data from a voxel, even if that voxel contains no actual signal. Thus, for all GLMsingle beta versions that do not include ridge regression (*b*1 - *b*3), users should avoid interpreting the GLM $R^2$ values, as they will be inflated across the brain and not provide a reliable index of signal-to-noise ratio (SNR). However, the ridge regression technique (as a direct consequence of its goal of optimizing cross-validated generalizability of the single-trial beta estimates) will tend to leave unperturbed the voxels that have good SNR and aggressively regularize the voxels with little or no SNR. As a consequence, the GLM $R^2$ values produced by ridge regression (*b*4) will be directly indicative of SNR, and therefore can be informative to the user.

## Potential limitations to consider when applying GLMsingle

GLMsingle relies on cross-validation to determine two key hyperparameters: (i) the number of nuisance regressors to use in the GLM as derived by applying PCA to data from the noise pool voxels; and (ii) the amount of ridge-regression shrinkage to apply for each voxel. Although the data-driven nature of the technique is one of its strengths (since it adapts to the characteristics of each dataset), it is also a potential limitation. First, a prerequisite for application of GLMsingle is the existence of at least some repeated trials in a given dataset. A dataset consisting only of experimental conditions with a single occurrence each cannot be used in conjunction with the cross-validated procedures for determining the optimal number of nuisance regressors and the voxel shrinkage fractions. Second, since data are invariably noisy, the determination of hyperparameters is subject to noise, and it is not guaranteed that hyperparameter estimates will be accurate in all possible data situations. It remains an open question for further investigation what the minimum data requirements are for reasonably accurate hyperparameter estimation.

Given the requirement of repeated discrete events, GLMsingle is not applicable to resting-state data, since they contain no explicit task structure. Similarly, GLMsingle is not suitable for experiments that involve continuous event structures – for example, movie watching, storytelling, dynamic exploration, game-playing – unless certain events within the task are coded as discrete, repeated instances. For example, the appearance on-screen of a particular character could be treated as a repeated "event" in constructing a design matrix. Or, as another example, certain words or parts of speech could be treated as "events" within a continuous auditory or linguistic experiment.

It is important to consider whether denoising comes at the potential cost of introducing bias (*Kay, 2022*). Considering each component of GLMsingle, we believe that the risk of bias is minimal for most use cases. First, considering the library-of-HRFs approach, we note that the conventional approach of using a fixed canonical HRF actually incurs more risk of biasing response estimates than does an approach that attempts to flexibly capture variations in HRFs. Nonetheless, we acknowledge that the library may not necessarily capture all HRF shapes, and this represents one possible source of bias (note that it is possible to derive a new HRF library tailored to a given dataset of interest). Second, considering the GLMdenoise procedure, we note that data-derived nuisance regressors are not blindly removed from the time-series data prior to modeling, as this would pose a clear risk of removing experimentally-driven signals, thereby leading to bias (*Liu et al., 2001*). Rather, by including both task-related regressors and nuisance regressors in the GLM, the model can appropriately partition variance between signal and noise sources. Third, considering ridge regression, we note that

shrinkage can be viewed as a form of temporal smoothing (as explained above). While shrinkage is indeed a source of bias, this should be concerning only for investigations where relative responses to nearby trials are of specific interest (e.g. studies of repetition suppression). For other investigations, and especially for experiments where condition ordering is pseudorandom, it is unlikely that this form of temporal regularization and its associated bias would lead to incorrect scientific inferences.

More generally, it is important to realize that GLMsingle uses limited and principled signal processing methods to improve the quality of BOLD signal estimates. It is not meant to – nor is able to – arbitrarily remove variability from a set of data, variability that may in fact be of interest to the researcher (e.g. trial-to-trial variations in response due to changes in behavioral state). Rather, GLMsingle is capable of modifying the data in only relatively constrained ways, for example, by damp-ening temporal instabilities in beta estimates (ridge regression) and by removing variance attributable to nuisance regressors (GLMdenoise). As we have demonstrated, applying GLMsingle to empirical data provides major substantive benefits with respect to downstream analytical outcomes.

## Online example scripts and tutorials

To enable easy adoption of GLMsingle, we provide extensive documentation and example scripts for common neuroimaging use-cases (glmsingle.org). Publicly available online resources include code implementation of GLMsingle in both MATLAB and Python, example scripts and notebooks, tech-nical documentation, and answers to frequently asked questions. The GLMsingle pipeline is designed to be easy to implement in different neuroimaging pipelines. The example scripts we provide illus-trate typical GLMsingle usage for both event-related and block designs. These scripts guide the user through basic calls to GLMsingle, using representative, small-scale example datasets. In addition, they provide helpful visualizations related to inspection and interpretation of intermediate outputs from the pipeline (e.g. optimal HRF indices and ridge regression shrinkage fractions). We hope these prac-tical resources facilitate the application of GLMsingle to existing and future neuroimaging datasets.

## Conclusion

Our results suggest that GLMsingle represents a methodological advancement that will help improve data quality across different fMRI designs. While improvements in MR hardware (e.g. magnetic field strength, RF coil, pulse sequences) and experimental design (e.g. optimized study design and trial distributions) may contribute to improved data quality, the benefits of GLMsingle demonstrated in this paper make clear that data processing techniques are another critical factor that can profoundly impact SNR and overall experimental power. As an analogy, we observe that the rapid (and annual) improvement in cell phone cameras has been driven in large part by advances in image analysis algo-rithms. As summarized by an Apple executive, "[while sensor quality has improved], increasingly, what makes incredible photos possible aren't just the sensor and the lens but the chip and the software that runs on it" (*Wilson, 2018*). We suggest that GLMsingle represents a similar advance in signal processing for fMRI.

## Materials and methods
### Description of GLMsingle
#### Inputs to GLMsingle

GLMsingle expects that input fMRI data have been pre-processed with motion correction at minimum, and ideally slice time correction as well. Additional common pre-processing steps such as compen-sation for spatial distortion, spatial smoothing, or registration to an anatomical space (or atlas space) are all compatible with GLMsingle without any complications. Detrending or high-pass filtering the time-series data is not necessary, as low-frequency fluctuations are modeled as part of the GLM fitting procedure. The input fMRI data can be supplied in either volumetric or surface format. Besides fMRI data, the other user-provided input to GLMsingle is an array of design matrices corresponding to each run of the time-series data, indicating the sequence of events that occurred during the runs. GLMsingle expects that these are matrices with dimensions (time x conditions), where each column corresponds to a single condition and consists of zeros except for ones indicating the onset times for that condition. Further details about data formats are provided in the online code repository.

## GLMsingle overview

GLMsingle consists of three main analysis components. The first component is the use of a library of hemodynamic response functions (HRFs) to identify the best-fitting HRF for each voxel. This simple approach for compensating for differences in hemodynamic timecourses across voxels (*Handwerker et al., 2004*) has several appealing features: it invariably provides well-regularized HRF estimates, and it is efficient and can be executed with reasonable computational cost. The second component is an adaptation of GLMdenoise to a single-trial GLM framework. GLMdenoise is a previously introduced technique (*Kay et al., 2013*) in which data-derived nuisance regressors are identified and used to remove noise from – and therefore improve the accuracy of – beta estimates. The third analysis component is an application of ridge regression (*Hoerl and Kennard, 1970*) as a method for dampening the noise inflation caused by correlated single-trial GLM predictors. To determine the optimal level of regularization for each voxel, we make use of a recently developed efficient re-parameterization of ridge regression called "fractional ridge regression" (*Rokem and Kay, 2020*).

## Derivation of the library of HRFs

The HRF library incorporated into GLMsingle was previously used for signal estimation in analyzing the Natural Scenes Dataset. Complete details on the derivation procedure for the HRF library can be found in the NSD dataset paper (*Allen et al., 2022*). In brief, empirically observed BOLD timecourses were subject to principal components analysis, projected onto the unit sphere, and parameterized using a path consisting of 20 regularly-spaced points through the area of greatest data density. The timecourses corresponding to the resulting set of 20 points were fit using a double-gamma function as implemented in SPM's spm_hrf.m, yielding a fixed library of 20 HRFs. This library is the default in GLMsingle, and was used for all analyses of the NSD, BOLD5000, and StudyForrest datasets described here. Future studies may seek to refine or expand the HRF library (e.g. by deriving it from a larger pool of subjects, or by restricting estimation to individual subjects).

## Cross-validation framework for single-trial GLMs

The GLMdenoise and ridge regression analysis components of GLMsingle both require tuning of hyperparameters (specifically, the number of nuisance regressors to include in GLM fitting and the regularization level to use for each voxel). To determine the optimal setting of hyperparameters, we use a cross-validation approach in which out-of-sample predictions are generated for single-trial beta estimates. Performing cross-validation on single-trial betas, as opposed to time-series data, simplifies and reduces the computational requirements of the cross-validation procedure. Note that because of cross-validation, although GLMsingle produces estimates of responses to single trials, it does require the existence of and information regarding repeated trials (that is, trials for which the experimental manipulation is the same and expected to produce similar brain responses). This requirement is fairly minimal, as most fMRI experiments are designed in this manner.

The first step of the cross-validation procedure is to analyze all of the available data using a generic GLM. In the case of GLMdenoise, this amounts to the inclusion of zero nuisance regressors; in the case of ridge regression, this amounts to the use of a shrinkage fraction of 1, which corresponds to ordinary least-squares regression. In both cases, the generic analysis produces a full set of unregularized single-trial betas (e.g. in one NSD session, there are 750 single-trial betas distributed across 12 runs, and in one BOLD5000 session, there are either 370 or 333 single-trial betas distributed across either 10 or 9 runs). The second step of the procedure is to perform a grid search over values of the hyperparameter (e.g., number of GLMdenoise nuisance regressors; ridge regression shrinkage fraction). For each value, we assess how well the resulting beta estimates generalize to left-out runs. By default, for all cross-validation procedures, GLMsingle implements the following leave-one-run-out routine: (1) one run is held out as the validation run, and experimental conditions that occur in both the training runs and the validation run are identified; (2) squared errors between the regularized beta estimates from the training runs and the unregularized beta estimates from the validation run are computed; (3) this procedure is repeated iteratively, with each run serving as the validation run, and errors are summed across iterations.

## GLMsingle algorithm

Having described the essential aspects of the estimation framework above, we now turn to the steps in the GLMsingle algorithm. GLMsingle involves fitting several different GLM variants. Each variant includes polynomial regressors to characterize the baseline signal level: for each run, we include polynomials of degrees 0 through $round(L/2)$ where $L$ is the duration in minutes of the run.

1. *Fit a simple ON-OFF GLM.* In this model, all trials are treated as instances of a single experimental condition, and a canonical HRF is used. Thus, there is a single "ON-OFF" predictor that attempts to capture signals driven by the experiment. The utility of this simple model is to provide variance explained ($R^2$) values that help indicate which voxels carry experimentally-driven signals.
2. *Fit a baseline single-trial GLM.* In this model, each stimulus trial is modeled separately using a canonical HRF. This model provides a useful baseline that can be used for comparison against models that incorporate more advanced features (as described below).
3. *Identify an HRF for each voxel.* We fit the data multiple times with a single-trial GLM, each time using a different HRF from the library of HRFs. For each voxel, we identify which HRF provides the best fit to the data (highest variance explained), and inherit the single-trial betas associated with that HRF. Note that the final model for each voxel involves a single chosen HRF from the library.
4. *Use GLMdenoise to determine nuisance regressors to include in the model.* We define a pool of noise voxels (brain voxels that have low ON-OFF $R^2$, according to an automatically determined threshold) and then perform principal components analysis on the time-series data associated with these voxels (separately for each run). The top principal components (each of which is a timecourse) are added one at a time to the GLM until cross-validation performance is maximized on-average across voxels. The inclusion of these nuisance regressors is intended to capture diverse sources of noise that may contribute to the time-series data in each voxel.
5. *Use fractional ridge regression to regularize single-trial betas.* With the nuisance regressors determined, we use fractional ridge regression to determine the final estimated single-trial betas. This is done by systematically evaluating different shrinkage fractions. The shrinkage fraction for a given voxel is simply the ratio between the vector length of the set of betas estimated by ridge regression and the vector length of the set of betas returned by ordinary least-squares estimation, and ranges from 0 (maximal regularization) to 1 (no regularization). For each voxel, in the context of a GLM that incorporates the specific HRF chosen for that voxel as well as the identified nuisance regressors, cross-validation is used to select the optimal shrinkage fraction.

The default behavior of GLMsingle is to return beta weights in units of percent signal change by dividing by the mean signal intensity observed at each voxel and multiplying by 100. To preserve the interpretability of GLM betas as percent signal change even after applying shrinkage via ridge regression, we apply a post-hoc scaling and offset on the betas obtained for each voxel in order to match, in a least-squares sense, the unregularized betas (shrinkage fraction equal to 1) obtained for that voxel.

To give a sense of the computational requirements of GLMsingle, we report here results for an example scenario. We ran the MATLAB version of GLMsingle with default parameters on the first NSD scan session for subj01 (1.8-mm standard-resolution version of the data). The scan session involved 750 trials and a data dimensionality of (81 voxels × 104 voxels × 83 voxels) = 699,192 voxels and (12 runs × 226 volumes) = 2712 time points. The code was run on an 32-core Intel Xeon E5-2670 2.60 GHz Linux workstation with 128 GB of RAM and MATLAB 9.7 (R2019b). The data were loaded in single-precision format, resulting in a base memory usage of 8.4 GB of RAM (the data alone occupied 7.6 GB). Code execution (including figure generation and saving results to disk) took 4.8 hr (average of 2 trials). The maximum and mean memory usage over the course of code execution was 38.0 GB and 18.5 GB of RAM, respectively.

## GLMsingle outputs

The default output from GLMsingle includes the different GLM beta estimates that are progressively obtained in the course of the algorithm (e.g. the single-trial betas with voxel-wise tailored HRFs; the single-trial betas incorporating GLMdenoise, etc.). The pipeline also outputs several metrics of interest, such as a map of the HRF indices chosen for different voxels, the $R^2$ values from the ON-OFF GLM, a map of the voxels identified as "noise", a summary plot of the cross-validation procedure

used to select the number of noise regressors, and a map of the amount of ridge regression shrinkage applied at each voxel. These outputs are displayed in a set of convenient figures.

## Flexibility of GLMsingle

Although GLMsingle provides default settings for the parameters that control its operation, the toolbox is flexible and allows the user to adjust the parameters if desired. Modifying the parameters allows the user to achieve a range of different behaviors, such as expanding the HRF library to include additional candidate HRFs; changing the maximum number of nuisance regressors tested during GLMdenoise (default is 10); modifying the range of shrinkage fractions evaluated for ridge regression (default is 0.05 to 1 in increments of 0.05); and running different flavors of GLM models that omit HRF fitting, GLMdenoise, and/or ridge regression. For complete documentation, please refer to the GLMsingle function descriptions and example scripts available at glmsingle.org.

## Application of GLMsingle to NSD and BOLD5000

In order to assess the efficacy of GLMsingle for large-scale fMRI datasets, we tested GLMsingle on the NSD (*Allen et al., 2022*) and BOLD5000 (*Chang et al., 2019*) datasets. Both datasets involve presentation of many thousands of natural images. NSD and BOLD5000 share an overlapping subset of stimuli from the Microsoft Common Objects in Context (COCO) database (*Lin et al., 2014*), enabling direct comparison between the brain responses observed in the two datasets. However, there are a number of differences between the datasets: they were collected at different field strengths, with different event timings, and at different spatial and temporal resolution. In addition, while NSD contains many repeated stimuli within each scan session, BOLD5000 contains very few. As such, processing BOLD5000 required grouping of input data across scan sessions to facilitate the cross-validation procedures used in GLMsingle. This challenging processing scheme with respect to image repetitions provides a strong test of the robustness of the GLMsingle technique.

### NSD dataset

For complete details of the NSD study, including scanning parameters, stimulus presentation, and experimental setup, refer to the *Methods* section of the corresponding dataset paper (*Allen et al., 2022*). In brief, a total of eight subjects participated in the NSD experiment, each completing between 30 and 40 functional scanning sessions. For the full experiment, 10,000 distinct images from the Microsoft COCO dataset were presented three times each over the course of 40 sessions. For computational convenience and to make comparisons across subjects easier, only the first 10 NSD sessions from subjects 1-4 were used for the analyses contained in this manuscript. Functional data were collected at 7T, with 1.8-mm isotropic resolution, and with a TR of 1.6 s. Each trial lasted 4 s, and consisted of the presentation of an image for 3 s, followed by a 1-s gap. A total of 12 NSD runs were collected in one session, containing either 62 or 63 stimulus trials each, for a total of 750 trials per session.

The fMRI data from NSD were pre-processed by performing one temporal resampling to correct for slice time differences and one spatial resampling to correct for head motion within and across scan sessions, EPI distortion, and gradient nonlinearities. This procedure yielded volumetric fMRI time-series data in subject-native space for each NSD subject. For the present study, we analyzed the standard-resolution pre-processed data from NSD which has 1.8-mm spatial resolution and 1.333-s temporal resolution (the time-series data are upsampled during pre-processing).

### BOLD5000 dataset

For complete details of the BOLD5000 study and methodology, refer to the corresponding dataset paper (*Chang et al., 2019*). A total of four subjects participated in the BOLD5000 dataset (CSI1-4). A full dataset contained 15 functional scanning sessions; subject CSI4 completed only nine sessions before withdrawing from the study. BOLD5000 involved presentation of scene images from the Scene UNderstanding (SUN; *Xiao et al., 2010*), COCO (*Lin et al., 2014*), and ImageNet (*Deng et al., 2009*) datasets. A total of 5254 images, of which 4916 images were unique, were used as the experimental stimuli. Of the 4916 distinct images, 112 were shown four times and one image was shown three times to each subject. Functional data were collected at 3T, with 2-mm isotropic resolution, and with a TR of 2 s. Each trial lasted 10 s, and consisted of the presentation of an image for 1 s, followed by a 9-s

gap. A total of either 9 or 10 runs were collected in one session, containing 37 stimulus trials each, for a total of either 333 or 370 trials per session.

The fMRI data from BOLD5000 were pre-processed using fMRIPrep (*Esteban et al., 2019*). Data pre-processing included motion correction, distortion correction, and co-registration to anatomy (for further details, please refer to the BOLD5000 dataset paper *Chang et al., 2019*). This yielded volumetric fMRI time-series data in subject-native space for each BOLD5000 subject.

Because GLMsingle requires condition repetitions in order to perform internal cross-validation procedures, and because BOLD5000 contains a limited number of within-session repetitions, we concatenated data from groups of 5 sessions together before processing using GLMsingle. To account for differences in BOLD signal intensity across different sessions, we applied a global rescaling operation to the data within each session to roughly equate the time-series mean and variance across the five sessions comprising one batch of data. Specifically, we first computed the global mean fMRI volume across all five sessions, and then, for each session, computed a linear fit between the mean volume from a single session and the global mean volume. This yielded a multiplicative scaling factor to apply to each session in order to roughly equate signal intensities across sessions.

## Applying GLMsingle to NSD and BOLD5000

We used GLMsingle to estimate single-trial BOLD responses in the NSD and BOLD5000 datasets. For NSD, GLMsingle was applied independently to each scan session. For BOLD5000, groups of five sessions were processed together, following the rescaling procedure described above. The default GLMsingle parameters were used for processing both NSD and BOLD5000, except that we evaluated up to 12 nuisance regressors in GLMdenoise (default: 10).

Four different versions of single-trial GLM betas were computed and saved. The first beta version (*b*1, AssumeHRF) is the result of Step 2 of the GLMsingle algorithm, and reflects the use of a canonical HRF with no extra optimizations. We treat these generic GLM outputs as a baseline against which beta versions are compared; estimating BOLD responses using a canonical HRF and ordinary least-squares (OLS) regression reflects an approach that has been commonly applied in the field of human neuroimaging. The second beta version (*b*2, FitHRF) is the result of Step 3, and reflects the result of voxel-wise HRF estimation. The third beta version (*b*3, FitHRF + GLMdenoise) is the result of Step 4, incorporating GLMdenoise, and the final beta version (*b*4, FitHRF + GLMdenoise + RR) arises from Step 5, and reflects the additional use of ridge regression. For comparisons between GLMsingle and Least-Squares Separate (LSS) signal estimation (*Figure 3*), four auxiliary beta versions were computed. LSS betas were compared to those estimated using fractional ridge regression in the scenario of using the canonical HRF (AssumeHRF + LSS vs. AssumeHRF + RR) and in the scenario of performing HRF optimization using the GLMsingle library (FitHRF + LSS vs. FitHRF + RR). Our validation analyses involved comparing optimized GLMsingle betas (*b*2, *b*3, *b*4) against those estimated using the baseline GLM approach (*b*1), and performing an eight-way comparison incorporating both *b*1-*b*4 and the four auxiliary beta versions used for comparisons with LSS. Prior to all analyses, the responses of each voxel were *z*-scored within each experimental session in order to eliminate potential nonstationarities arising over time, and to equalize units across voxels.

## Applying GLMsingle to the StudyForrest music-listening dataset

As a further test of the general applicability of GLMsingle, we repeated the above procedures using data from the music-listening component of StudyForrest (*Hanke et al., 2015*). This dataset measures brain responses as subjects listen to 25 total musical clips from 5 genres, with 8 repetitions per condition per subject. We analyzed the group of 16 subjects from this dataset for whom all functional data files were available within the online data repository. Each stimulus was a 6-s excerpt from the middle of a distinct musical piece, for each of five different musical genres: Ambient, Country, Heavy Metal, 50s Rock'n'Roll, and Symphonic. All trials had 4, 6, or 8 s of delay (no audio, white fixation cross) after each musical stimulus, with the order of delays randomized within a run. The 25 stimuli were identical across runs and presented exactly once per run. Each subject completed 1 session of the experiment, which consisted of 8 runs. BOLD data were distortion-corrected and anatomically aligned to a per-subject BOLD template image prior to analysis (see *Hanke et al., 2015* for acquisition and pre-processing details). GLMsingle was applied to the data from each subject using default hyperparameter settings.

## Assessing the impact of GLMsingle

### Analysis of optimal HRF indices

We sought to analyze the structure of HRF indices across visually-responsive cortex of a representative subject (NSD subj01). To identify voxels that were responsive to experimental stimuli, we examined the $R^2$ output from the ON-OFF GLM that is computed in the course of deriving signal estimates for each session – this value reflects the variance explained in the observed BOLD time series by a single predictor containing all experimental events coded jointly. To derive a single metric of signal quality, we computed the session-averaged ON-OFF $R^2$ value for all voxels within the nsdgeneral mask. These values are plotted across three different threshold levels in *Figure 2—figure supplement 1a*. *Figure 2—figure supplement 1b* shows the optimal HRF indices derived from the first scan session from NSD subj01. To estimate the consistency of optimal HRF indices in this subject, we identified the group of nsdgeneral ROI voxels corresponding to each threshold level (rows, *Figure 2—figure supplement 1*) and computed the Pearson correlation between the pattern of HRF indices identified for each of the 10 sessions that were analyzed (*Figure 2—figure supplement 1c*).

### Computing voxel test-retest reliability

To compute reliability in NSD and BOLD5000, we repeated the following procedure for each beta version. We first extracted the betas from trials that correspond to repetitions of the same stimuli (NSD: 3 instances per stimulus; BOLD5000: 4 instances for subjects CSI1-3, and 3 for CSI4). For each voxel, this yielded a matrix of dimensions (repetitions x images). To compute reliability, Pearson correlation was computed between the average voxel response profiles for each possible unique split-half of the data. Therefore, in the case of 4 available repetitions, the reliability for a voxel was the average of 3 correlation values, with image repetitions grouped as follows: *corr*(*mean*(1,2) vs. *mean*(3,4)); *corr*(*mean*(1,3) vs. *mean*(2,4)); *corr*(*mean*(1,4) vs. *mean*(2,3)). In the case of three repetitions, the reliability was the average of: *corr*(*mean*(1,2) vs. (3)); *corr*(*mean*(1,3) vs. (2)). In the Study-Forrest data, the reliability of response estimates in each voxel was computed using the following test-retest reliability procedure: the available repetitions (8) were divided into all possible unique split-halves (70 possibilities); for each split, the 25-valued response profiles for each group of 4 repetitions were averaged; the Pearson correlation between the response profiles from the two groups was computed; and correlation values were averaged across the 70 possible splits, yielding a single reliability value per voxel. Although Pearson correlation makes certain distributional assumptions, we suspect that the basic trends in our results would be unchanged were we to use other metrics for quantifying reliability (e.g. Spearman rank correlation).

### ROI analysis within visual cortex

To summarize reliability outcomes for each beta version, we used a liberal mask containing voxels in visual cortex. Specifically, we used the nsdgeneral ROI from the NSD study, which was manually drawn on fsaverage to cover voxels responsive to the NSD experiment in the posterior aspect of cortex (*Allen et al., 2022*). To achieve a common reference ROI in volumetric space for each subject, we first transformed the nsdgeneral ROI to MNI space, and then mapped this ROI from MNI space to the space of each subject in NSD and each subject in BOLD5000.

### Composite voxel reliability scores

In comparing different beta versions output by GLMsingle, we sought to understand whether the optimizations tended to affect all voxels equally, or whether the impact was mediated by voxel reliability. We therefore measured how different beta versions differed in our key outcome metrics (e.g. mean voxel reliability) as a function of the reliability of included voxels. To achieve fair comparisons, we ensured that the same groups of voxels were compared at each reliability threshold across beta versions. We achieved this by computing composite voxel reliability scores: the mean reliability value in each voxel over beta versions $b$1-$b$4. We then subselected groups of voxels by applying varying threshold levels to the composite reliability scores. For analyses incorporating the 4 auxiliary beta versions, composite reliability scores were computed as the mean across all 8 beta versions.

## Beta version comparisons as a function of voxel reliability

To quantify the performance of different beta versions as a function of voxel reliability, composite scores were thresholded at increasing values (from Pearson $r = -0.2$ to 0.6, in steps of 0.05) to determine the included voxels at each reliability level. At each threshold, we computed the difference between the reliability achieved by a given beta version and the composite reliability (i.e. the average across beta versions). This difference was averaged across voxels, producing traces that reflect the relative quality of data from each beta version compared to the group average, across different levels of voxel reliability (*Figure 2b*).

## Out-of-sample reliability analysis

GLMsingle makes use of all of the data that it is presented with, via a series of internal cross-validation operations. As such, there is some degree of dependence between runs. Note that this does not pose a significant "circularity" problem with respect to downstream analyses, as GLMsingle has no access to any scientific hypotheses and it is unlikely that GLMsingle could bias the single-trial beta estimates in favor of one hypothesis over another. However, when the primary analysis outcome is to establish that responses to the same condition are reliable across trials (e.g. *Figures 2 and 3*), then that outcome is exactly what the GLMsingle algorithm is trying to achieve during hyperparameter selection. For a stringent quantification of reliability, we performed additional analyses in which quantification of reliability was restricted to responses estimated in completely independent calls to GLMsingle (*Figure 3b*). Specifically, we identified all instances where a condition was repeated within the same partition of data processed by GLMsingle (partition size: 1 session for NSD, 5 sessions for BOLD5000), and removed these instances from the calculation of reliability. The results show that even with strict separation, the patterns of results are essentially the same.

## Comparison to LSS

Least-Squares Separate (LSS) is a popular technique for robust signal estimation in rapid event-related designs (*Mumford et al., 2012*; *Mumford et al., 2014*; *Abdulrahman and Henson, 2016*). The LSS procedure fits a separate GLM for each stimulus, where the trial of interest is modeled as one regressor, and all other (non-target) trials are collapsed into a second regressor. An implementation of LSS is included in the GLMsingle toolbox.

## Analysis of reliability in StudyForrest

*Figure 4* shows the results of analyses that seek to validate the efficacy of GLMsingle using data from the music-listening task in StudyForrest. We examine differences in the test-retest reliability (Pearson $r$) of active voxels between $b1$ and $b4$ in *Figure 4a*. Active voxels are identified using a thresholding procedure applied to the ON-OFF $R^2$ values, and only voxels that are active in response to experimental stimuli (ON-OFF $R^2 > 5$) are plotted. To compute relative differences in mean reliability between beta versions $b1$ - $b4$ (*Figure 4c–d*), we used an identical procedure to that used for NSD and BOLD5000 (*Figure 2b*). Traces in *Figure 4c* indicate the mean (+/- SEM) across the 16 available subjects, whose data are plotted individually in *Figure 4d*.

## Analysis of temporal autocorrelation

A commonly used strategy to increase fMRI statistical power is to increase the number of experimental trials by allowing them to be presented close together in time. However, given the sluggish nature of BOLD responses and the existence of temporal noise correlations, this strategy tends to yield correlations in GLM beta estimates for nearby trials (*Mumford et al., 2014*; *Olszowy et al., 2019*; *Woolrich et al., 2001*; *Kumar and Feng, 2014*). In general, we expect that such correlations are largely artifactual and unwanted. Given that GLMsingle attempts to reduce noise levels, we sought to explore whether GLMsingle has a noticeable impact on temporal autocorrelation.

For each beta version, the following procedure was used to assess the degree of temporal autocorrelation in the data. For visual cortex data from each experimental session (nsdgeneral ROI, *Allen et al., 2022*), we computed the Pearson correlation between the spatial response patterns from each pair of trials in the session, yielding a representational similarity matrix (RSM) where the temporal ordering of trials is preserved. This process was repeated for all sessions, yielding a total of 10 RSMs for each NSD subject and 15 RSMs for each BOLD5000 subject (9 for subject CSI4, who did not

complete the full study). To assess autocorrelation in the data – relationships arising due to temporal proximity of different trials – we then took the average of all RSMs within each dataset. Note that in both NSD and BOLD5000, the order of stimulus presentation was essentially unstructured (pseudorandom). Thus, in terms of signal content (stimulus-driven responses in the absence of noise), we expect that trials should be uncorrelated, on average, and that any non-zero correlations are indicative of the effects of noise that persist following GLM fitting. The extent to which non-zero $r$ values extend forward in time from the RSM diagonal indicates the timescale of the noise effects in a given beta version.

## Computing the autocorrelation function

To quantify the degree of temporal autocorrelation, we computed a mean autocorrelation function using the dataset-averaged RSM for each beta version (*Figure 5*). For a given RSM, we computed the average similarity value between all trials $k$ and $k + x$, where $x$ varies from 1 to $n$, where $n$ is the dimensionality of the RSM. Intuitively, at $x = 1$, $autocorr(x)$ equals the average of all values falling 1 index below the diagonal of the RSM; at $x = 5$, it equals the average of all values falling 5 indices below the diagonal, etc. This procedure outputs a succinct summary of the average correlation in neural response between all pairs of time-points within a session, allowing for easy comparison between the beta versions in a single plot (*Figure 5*, right-most column). The theoretical desired outcome is $autocorr(x) = 0$; thus, beta versions whose autocorrelation functions are "flatter" (e.g. less area under the curve) presumably contain more accurate GLM estimates. Because the temporal interval between trials differed between NSD (4 s) and BOLD5000 (10 s), we express the autocorrelation functions in terms of seconds post-stimulus for plotting, to allow for straightforward comparison between the datasets.

## Effect of voxel reliability on temporal autocorrelation

The effect of temporal autocorrelation in GLM betas may vary depending on the relative responsiveness of different voxels to the experimental stimuli. As such, we repeated the autocorrelation analyses several times, varying the expanse of voxels that were included. We again relied on the aggregate reliability scores (computed previously) as a measure of voxel quality, which are the average voxel reliabilities taken across all the beta versions under consideration. This avoids biasing the voxel selection procedure. In *Figure 5*, we compare temporal autocorrelation trends arising from analysis of voxels at two different reliability thresholds ($r = 0$ and $r = 0.3$).

## Analysis of between-subject representational similarity

Another way to assess the quality of beta estimates is to examine the similarity of BOLD response estimates across subjects. The underlying logic is that noise is expected to be stochastic in the data acquisition for each subject, and thus, should on average increase the dissimilarities of BOLD response estimates across subjects. A method that accurately removes noise would then be expected to increase the similarity of BOLD responses across subjects. To quantify response similarity, we use representational similarity analysis (RSA), a commonly used approach in systems and cognitive neuroscience (*Kriegeskorte et al., 2008*; *Nili et al., 2014*; *Diedrichsen and Kriegeskorte, 2017*; *Kaniuth and Hebart, 2022*).

## Analysis of image-level MVPA decoding accuracy

Multivoxel pattern analysis (MVPA) investigates the information contained in distributed patterns of neural activity to infer the functional role of brain areas and networks. Pattern decoding tools such as MVPA have been deployed extensively in systems and cognitive neuroscience to study the function of neural ROIs (*Haxby et al., 2001*; *Norman et al., 2006*; *Naselaris et al., 2011*; *Charest et al., 2018*). To further assess the practical impact of GLMsingle, we tested the efficacy of MVPA decoding using the different beta versions output by the toolbox.

We implemented a challenging "one-vs.-many" decoding task to assess whether data quality was sufficiently high to characterize the distinct neural patterns associated with individual naturalistic images in the NSD and BOLD5000 datasets. Within each dataset, we identified the set of images that all subjects viewed at least three times, and then performed multiclass linear support vector machine (SVM) decoding via leave-one-repetition-out cross-validation. In NSD, a total of 82 classes

were used, representing the images that overlapped across the 10 available sessions from subj01-04. In BOLD5000, the subset of these 82 stimuli overlapping between all subjects of both datasets were used (a total of 20 classes). We then assessed the degree to which relative differences in decoding accuracy between $b1$ and $b4$ changed depending on the reliability of the included voxels. We conducted the above decoding procedure iteratively, each time increasing the voxel reliability inclusion threshold for data within the nsdgeneral ROI (range $r$ = 0 to 0.35). BOLD5000 subject CSI4, having completed only 9 of 15 experimental sessions, was excluded from MVPA procedures due to insufficient stimulus repetitions.

## Multidimensional scaling

To gain insight into the representational changes due to GLMsingle that may support improvements in MVPA decoding, we performed multidimensional scaling (MDS) over repetition-averaged NSD betas from a baseline GLM ($b1$) and the final betas from GLMsingle ($b4$), within the nsdgeneral ROI of an example subject (NSD subj01). In *Figure 7b*, we compare the 2-dimensional MDS embeddings between these beta versions, coloring COCO stimuli based on whether they contain animate or inanimate objects according to the image annotations.

# Acknowledgements

Collection of the NSD dataset was supported by NSF CRCNS grants IIS-1822683 (to KK) and IIS-1822929 (to Thomas Naselaris). We thank N Blauch, A Wang, E Aminoff, and R River for helpful discussions. We thank R Gau for help with the online GLMsingle code repository.

# Additional information

## Funding

| Funder | Grant reference number | Author |
| --- | --- | --- |
| National Science Foundation | IIS-1822683 | Kendrick N Kay |

The funders had no role in study design, data collection and interpretation, or the decision to submit the work for publication.

## Author contributions

Jacob S Prince, Conceptualization, Resources, Data curation, Software, Formal analysis, Validation, Investigation, Visualization, Methodology, Writing – original draft, Writing – review and editing; Ian Charest, Resources, Software, Validation, Methodology; Jan W Kurzawski, Resources, Software, Visualization, Writing – review and editing; John A Pyles, Conceptualization, Resources, Data curation, Supervision; Michael J Tarr, Conceptualization, Data curation, Supervision, Project administration, Writing – review and editing; Kendrick N Kay, Conceptualization, Resources, Data curation, Software, Formal analysis, Supervision, Funding acquisition, Validation, Investigation, Visualization, Methodology, Writing – original draft, Project administration, Writing – review and editing

## Author ORCIDs

Jacob S Prince ⓘ http://orcid.org/0000-0001-6169-9503
Ian Charest ⓘ http://orcid.org/0000-0002-3939-3003
Jan W Kurzawski ⓘ http://orcid.org/0000-0003-2781-1236
John A Pyles ⓘ http://orcid.org/0000-0002-7627-0504
Michael J Tarr ⓘ http://orcid.org/0000-0003-4724-1744
Kendrick N Kay ⓘ http://orcid.org/0000-0001-6604-9155

## Ethics

This paper uses previously published data from human subjects. For the Natural Scenes Dataset, informed written consent was obtained from all participants, and the experimental protocol was approved by the University of Minnesota institutional review board. For BOLD5000, participants all provided written informed consent and were financially compensated for their participation; all

procedures followed the principles in the Declaration of Helsinki and were approved by the Institutional Review Board of Carnegie Mellon University. For the StudyForrest dataset, all data acquisitions were jointly approved by the ethics committee of the Otto-von-Guericke University of Magdeburg, Germany. Subjects were fully instructed about the nature of the study and provided informed consent beforehand.

## Decision letter and Author response
Decision letter https://doi.org/10.7554/eLife.77599.sa1
Author response https://doi.org/10.7554/eLife.77599.sa2

## Additional files

### Supplementary files
• Transparent reporting form

### Data availability
The experimental data used in this paper are all previously published and available online at https://naturalscenesdataset.org, https://doi.org/10.18112/openneuro.ds001499.v1.3.1, and https://doi.org/10.18112/openneuro.ds000113.v1.3.0. The GLMsingle toolbox is available on GitHub at https://github.com/cvnlab/GLMsingle, (copy archived at swh:1:rev:3a1a580eae6bc7e221cbe01101b-e3c669687dc6a). Code used for the analyses demonstrated in this paper is publicly available on GitHub at https://github.com/jacob-prince/GLMsingle_paper, (copy archived at swh:1:rev:08cdd-25e4fd96f545056e516ec1910ab749e1887). Source data files linked to the results plotted in Figures 2-7 and Figure 2—figure supplement 1 are available in an OSF repository at https://osf.io/d4p98.

The following dataset was generated:

| Author(s) | Year | Dataset title | Dataset URL | Database and Identifier |
|---|---|---|---|---|
| Kay K, Kurzawski J, Charest I, Prince JS | 2022 | GLMsingle | https://doi.org/10.17605/OSF.IO/D4P98 | Open Science Framework, 10.17605/OSF.IO/D4P98 |

The following previously published datasets were used:

| Author(s) | Year | Dataset title | Dataset URL | Database and Identifier |
|---|---|---|---|---|
| Hanke M, Baumgartner FJ, Ibe P, Kaule FR, Pollmann S, Speck O, Zinke W, Stadler J | 2018 | Forrest Gump | https://doi.org/10.18112/openneuro.ds000113.v1.3.0 | OpenNeuro, 10.18112/openneuro.ds000113.v1.3.0 |
| Chang N, Pyles J, Prince J, Marcus A, Gupta A, Tarr M, Aminoff E | 2019 | BOLD5000 | https://doi.org/10.18112/openneuro.ds001499.v1.3.1 | OpenNeuro, 10.18112/openneuro.ds001499.v1.3.1 |
| Allen EJ, St-Yves G, Wu Y, Breedlove JL, Prince JS, Dowdle LT, Nau M, Caron B, Pestilli F, Charest I, Hutchinson JB, Naselaris T, Kay K | 2021 | Natural Scenes Dataset | https://naturalscenesdataset.org/ | University of Minnesota, naturalscenesdataset |

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
