## [Editor Report]

This important work provides the field of human neuroimaging with a new method to estimate single-trial fMRI responses. The authors provide compelling evidence that their GLMsingle method goes beyond the current state of the art and leads to more reliable estimates. Therefore, this tool will be of interest to researchers using human neuroimaging to study neural responses in condition-rich designs, as is increasingly common in cognitive neuroscience experiments.

---

## [Decision Letter]

**Decision letter after peer review:**

Thank you for submitting your article "GLMsingle: a toolbox for improving single-trial fMRI response estimates" for consideration by *eLife*. Your article has been reviewed by 3 peer reviewers, including Peter Kok as Reviewing Editor and Reviewer #1, and the evaluation has been overseen by Floris de Lange as the Senior Editor. The following individual involved in the review of your submission has agreed to reveal their identity: Benjamin Turner (Reviewer #3).

Essential revisions:

1) Including more practical guidelines on implementation would strengthen the manuscript and ease the implementation for users. This especially pertains to the features the user needs to flag when running the toolbox (HRF estimation, noise regressors, ridge regression). The specific implementation is left up to the users and the authors mention that this should depend on one's experimental goals, but do not provide more concrete guidelines. From the manuscript it seems that including all of the features works best, but this is based on two condition-rich experiments that may differ a lot from a user's dataset. Therefore it would be useful to walk the user through potential considerations for each of those features, or consider the most common uses of the toolbox (e.g. condition-rich designs, repetition-suppression studies, looking at within-session learning effects etc.).

2) There are some examples where the authors provide guidelines, but this could be strengthened. For instance, they discuss that the use of ridge regression could bias temporally adjacent trials to be more similar in magnitude – so they caution against using this feature for studies specifically interested in the relative responses of neighbouring trials (e.g. looking into preparation-execution in motor studies, repetition suppression-type of designs, etc.). But from Figure 4 there seems an advantage when including ridge regression in addition to denoising and fitting HRF, leading to a further reduction of the temporal autocorrelation between nearby trials. So, a reader might take-away that this is the least biased estimate of neighbouring trials. But mightn't it also destroy 'real' (neural) autocorrelation between trials, due to e.g. stimulus-specific adaptation and serial dependence? What exactly would the authors suggest then for designs where estimation of subsequent trials (e.g. repetition suppression or serial dependence) is of primary interest?

3) The authors use cross validation to determine the number of nuisance regressors to add in the model. Thus, any variability in responses to a single condition is considered to be 'noise'. How might this influence a potential use of single-trial estimates to assess brain-behaviour correlations (e.g., differences in behavioural responses to a single condition), or within-session learning conditions? For such uses, would the authors suggest instead using LSS or a subset of their features in GLMsingle (i.e. not using GLMdenoise)?

4) More generally, it would be ideal to see somewhere addressed the idea that variability is not always noise. You do mention repetition-suppression at one point, which is a clear example of this, but non-ergodicity as well as individual differences are further examples. There is no need to change the aims of the toolbox, which are clear and reasonable, but this somewhat tangential issue should at least be alluded to.

5) In the results, using a fixed HRF leads to drastically lower performance on a variety of subsequent measures compared to fitting an HRF to each voxel, especially as regards to β map test-retest reliability (Figure 2-3). Have the authors ensured that the HRF chosen is the most appropriate one for the region of interest? In other words, is the chosen HRF also the one that most voxels are fitted in the flexible option? It should be possible to quantify whether there is substantial dissimilarity in the chosen HRF from voxel to voxel. Since the HRFs span an equidistant arc, it would be expected that HRFs at opposite ends of the set are maximally dissimilar. Since the HRF has a biological interpretation, if it were frequently the case that neighboring voxels had dissimilar HRFs, this would be concerning.

6) It is a very small effect, but it would be interesting if the authors could speculate on the cost imposed by GLMdenoise in the very most-reliable voxels. Is this an artifact of the relatively small number of voxels that surpass this threshold? Or is there a chance the GLMdenoise step is removing signal? This refers to the rightmost point in the left plot of 3A (solid purple vs red; green vs orange). This is evident again in Figure 5B intra-NSD plot in the non-monotonicity from b2 to b3 for higher thresholds, and again when the b2 and b3 lines in Figure 6A (NSD) cross at r=0.25. Given that this does not seem to happen at all for BOLD5000, it is probably just an artifact, but seems nonetheless interesting enough for the authors to double-check whether there is any other explanation apparent in the data.

7) The benefit of b4 vs. b1 seems much larger in the NSD dataset than in the BOLD5000 dataset (Figure 2A). Is this because GLMsingle was initially optimised for the NSD dataset, or is there a different reason for this? The authors mention the fact that were fewer stimulus repetitions in BOLD5000 – but isn't that exactly the scenario for which GLMsingle was intended? Could it be due to the longer ITIs in BOLD5000?

8) The two datasets GLMsingle was tested on did not have a jittered intertrial interval (although the second one had a quite long (9s) intertrial interval). So, it remains to be seen whether there are also such large improvements when applying this method to a design with jittered intervals.

9) It would also be useful to include some intermediate results for the interested reader. As an example for the two chosen dataset, it could be instructional to know how many different hrf functions were obtained using FitHRF, how the ridge regression affects shrinkage of betas etc. The provided example in the toolbox (Python / Matlab) serves well to explore some of these intermediary steps, but some of these could also be explained in further detail as supplementary material. This would have a didactic purpose, informing the reader more about the process under the hood rather than just how the choices influence final estimates of betas.

10) Some relevant information on the amount of data from the two datasets could be explained in their Results section, specifically including number of conditions, repetitions per condition, and functional runs. It is not so straightforward to figure out this from the methods section given that the authors provide information on the datasets themselves, and then also the amount of the data used. Having some metrics in the main text would help to orient the reader and more easily allow a comparison of the results to the types of designs readers may be considering.

11) Researchers who study reliability often will complain about the use of Pearson correlation in that context. For completeness, the authors might want to at least look into this debate and decide whether it is worth addressing in the manuscript.

---

## [Author Response]

Essential revisions:1) Including more practical guidelines on implementation would strengthen the manuscript and ease the implementation for users. This especially pertains to the features the user needs to flag when running the toolbox (HRF estimation, noise regressors, ridge regression). The specific implementation is left up to the users and the authors mention that this should depend on one's experimental goals, but do not provide more concrete guidelines.

We agree that the manuscript would benefit from more extensive guidelines regarding the implementation of GLMsingle.

We have added a substantial “Implementation Guidelines” section to the Discussion. This section comments on major implementational choices that users may face when deciding how to apply GLMsingle to their own experimental paradigms. We include guidelines for the setup of design matrices and fMRI data for input to GLMsingle, and description of several input parameters that can enable, disable, or modify particular routines within GLMsingle (e.g. whether HRF fitting is enabled, the extent of the noise pool, the range of regularization levels tested during ridge regression). Beyond the concrete implementation tips, we also provide higher-level discussion to address key concerns or curiosities that users may have (these are designed based on our own personal interactions with many different researchers with whom we have communicated regarding GLMsingle). These additions to the manuscript should strengthen users’ understanding of the subroutines within GLMsingle and their rationales.

From the manuscript it seems that including all of the features works best, but this is based on two condition-rich experiments that may differ a lot from a user's dataset.

Our results indeed suggest that including all subroutines within GLMsingle yields the best outcomes in NSD and BOLD5000. We feel this to be reasonably strong evidence that each component of GLMsingle may contribute unique added benefit to the quality of signal estimates across a range of datasets, given the diversity of characteristics between the two datasets (e.g., different field strength, spatial and temporal resolution, behavioral tasks, event structure, stimuli). Still, we do acknowledge that, as large-scale, condition-rich datasets, NSD and BOLD5000 do differ from typical fMRI datasets. Thus, we agree that it would be additionally informative to demonstrate results on a new dataset that is further distinct from NSD and BOLD5000.

In the revised manuscript, we have conducted a further test of the generality of GLMsingle, using a third dataset consisting of 16 subjects performing an auditory task with 25 conditions (StudyForrest music-listening task). This dataset differs from NSD and BOLD5000 in three key ways: the task modality (auditory), the more “block” nature of the stimulus duration (6-s of auditory stimulation compared to 3-s and 1-s of visual stimulation in NSD and BOLD5000, respectively), and the use of a jittered inter-stimulus interval (see Essential Revision 8 below). As in NSD and BOLD5000, we observe substantial improvements in reliability via GLMsingle, and again, each individual component of GLMsingle confers added benefit in reliability compared to the baseline GLM (see Figure 4). These findings, arising from a smaller-scale dataset that may more closely reflect typical studies conducted by fMRI users, strengthen the evidence for the general applicability of GLMsingle.

(In addition, the example scripts provided on the GitHub repository show the effectiveness of GLMsingle on yet an additional dataset consisting of a conventional 10-condition category functional localizer. Of course, this is outside of the manuscript proper due to space reasons, but we just wanted to mention this to the reviewers and editors.)

Changes to the revised manuscript regarding the inclusion of the StudyForrest dataset can be found in the Introduction (paragraph 6); the Results section (under "GLMsingle improves the reliability of beta estimates" – see paragraph 9 of that subsection and Figure 4); and, in the Materials and methods section (under "Applying GLMsingle to the StudyForrest music-listening dataset").

Therefore it would be useful to walk the user through potential considerations for each of those features, or consider the most common uses of the toolbox (e.g. condition-rich designs, repetition-suppression studies, looking at within-session learning effects etc.).

We agree that adding further implementation guidelines regarding the main subroutines within GLMsingle (HRF fitting, GLMdenoise, and ridge regression) would be helpful to users, as well as recommendations regarding common use cases.

We have substantially expanded the Discussion section to provide additional detail on these topics.

2) There are some examples where the authors provide guidelines, but this could be strengthened. For instance, they discuss that the use of ridge regression could bias temporally adjacent trials to be more similar in magnitude – so they caution against using this feature for studies specifically interested in the relative responses of neighbouring trials (e.g. looking into preparation-execution in motor studies, repetition suppression-type of designs, etc.). But from Figure 4 there seems an advantage when including ridge regression in addition to denoising and fitting HRF, leading to a further reduction of the temporal autocorrelation between nearby trials. So, a reader might take-away that this is the least biased estimate of neighbouring trials. But mightn't it also destroy 'real' (neural) autocorrelation between trials, due to e.g. stimulus-specific adaptation and serial dependence? What exactly would the authors suggest then for designs where estimation of subsequent trials (e.g. repetition suppression or serial dependence) is of primary interest?

We acknowledge that studies seeking to analyze responses to trials that are close together in time pose challenges with respect to signal estimation, and users should indeed be mindful of how they analyze data using GLMsingle in these contexts. Furthermore, we point out that signal estimation for nearby trials is a general challenge that is not specific to GLMsingle: for example, ordinary least-squares estimates also suffer from statistical issues (e.g., high negative correlation in the estimation error associated with nearby trial estimates).

As the reviewers note, GLMsingle (perhaps surprisingly) can mitigate temporal autocorrelation in the spatial β maps between neighboring trials, which is presumably beneficial for most use cases that seek to disentangle the underlying brain responses to nearby trials. As noted in the manuscript, users should be aware that the use of ridge regression will lead to some amount of temporal smoothing of β estimates for nearby trials. When the critical hypotheses depend on the neural responses to trials that are directly after one another in time, users may choose to disable the ridge regression component of GLMsingle (to avoid any effects of temporal smoothing). But there is no guarantee that doing so will improve the accuracy of signal estimation – the trial-wise estimates associated with neighboring events will still have high levels of noise and exhibit statistical dependencies (correlations).

We think that there is no “silver bullet” for resolving the tricky issues associated with signal estimation when neighboring trials are of interest. The optimizations within GLMsingle are conservative by design (e.g., a single, constant, cross-validated ridge regression hyperparameter is applied per voxel over the entire set of trials passed to the function), and it is possible that the best outcomes may still be achieved with all components of GLMsingle enabled, even when close-by responses are of interest. This is an interesting issue that could be the subject of future, focused investigations.

We have added text to the Discussion addressing the issue of scenarios where estimates to neighboring trials are of particular interest. As an additional resource to users, we elaborate on these issues at length in the GLMsingle FAQ (https://glmsingle.readthedocs.io/en/latest/wiki.html#faq).

3) The authors use cross validation to determine the number of nuisance regressors to add in the model. Thus, any variability in responses to a single condition is considered to be 'noise'. How might this influence a potential use of single-trial estimates to assess brain-behaviour correlations (e.g., differences in behavioural responses to a single condition), or within-session learning conditions? For such uses, would the authors suggest instead using LSS or a subset of their features in GLMsingle (i.e. not using GLMdenoise)?

The reviewers are correct that implicit in computing cross-validated variance explained is the assumption that the same stimulus should generate the same brain response over repetitions. GLMsingle seeks to find the set of hyperparameters that results in the most replicable β estimates to repeated conditions within the chunk of data being processed. It is therefore reasonable to be concerned about the way the cross-validated components of GLMsingle would interact with analyses where different brain (and/or behavioral) responses are expected in response to condition repetitions.

Users who want to be extremely cautious may indeed want to disable the GLMdenoise and ridge regression components of the pipeline. But for most applications it seems unlikely that the nature of these regularizations would have undesirable effects on downstream analyses. For example, the purpose of cross-validation in GLMdenoise is merely to select a single scalar value dictating the number of PC noise regressors to include in the model – these PCs are derived from a pool of voxels that are not responsive to experimental stimuli, and it is unlikely that the inclusion (or exclusion) of particular PC regressors capturing global sources of noise would have the potential to destroy “true” signal that relates to fine-grained differences in neural responses to repeated conditions. As another example, GLMsingle uses ridge regression to select a single fractional regularization level for a given voxel – it is difficult to see how this could fundamentally influence a hypothesis pertaining to the full diversity of trial-wise responses obtained from a voxel. Thus, in short, there are strong theoretical reasons to believe that the outputs of GLMsingle are suitable for cases where one seeks to establish results that pertain to single trials (e.g. brain-behavioral correlations).

(Regarding the reviewers’ mention of LSS, we would not suggest substituting in least-squares separate (LSS) as an alternative to the ridge regression component of GLMsingle. LSS imposes more heavy-handed regularization on β estimates, and will corrupt β estimates when signal strength is high.)

The specifics of the cross-validation procedure and the general philosophy of GLMsingle with respect to variability and noise are discussed in the new Discussion section that has been added to the manuscript.

4) More generally, it would be ideal to see somewhere addressed the idea that variability is not always noise. You do mention repetition-suppression at one point, which is a clear example of this, but non-ergodicity as well as individual differences are further examples. There is no need to change the aims of the toolbox, which are clear and reasonable, but this somewhat tangential issue should at least be alluded to.

Here the reviewers raise the important point that variability is not always noise. As discussed in Essential Revision 3, we emphasize that GLMsingle is only capable of modifying the data in relatively constrained ways, for example by dampening temporal instabilities in β estimates (ridge regression), or removing bits of the signal that are contained in the GLMdenoise nuisance regressors. GLMsingle is not meant to – nor is able to – delete substantial amounts of “real” variability from the data. (In more intuitive terms: the cross-validation procedures are not bluntly taking all repetitions of a given condition and setting their betas to the mean across trials.) The goal of GLMsingle is to use limited and principled signal processing methods to improve the quality of BOLD signal estimates. And, as we have shown in the manuscript, doing so induces major benefits on downstream analytical outcomes.

Same as for Essential Revision 3, above.

5) In the results, using a fixed HRF leads to drastically lower performance on a variety of subsequent measures compared to fitting an HRF to each voxel, especially as regards to β map test-retest reliability (Figure 2-3). Have the authors ensured that the HRF chosen is the most appropriate one for the region of interest? In other words, is the chosen HRF also the one that most voxels are fitted in the flexible option?

As the reviewers note, we observe substantial improvement in the quality of signal estimates arising from the use of a custom HRF tailored by voxel. We interpret the reviewers’ query as referring to the issue of whether an assumed canonical HRF used in a given analysis is roughly matched to the general “true HRFs” associated with the voxels in a given region. This is an interesting issue. We point out that the vast majority of fMRI analyses being conducted in cognitive neuroscience use “off-the-shelf” HRFs that are not tailored at all to any given subject or region of interest. The assume-HRF analysis that we demonstrate in the paper is in line with this spirit.

The reviewers’ point leads us towards an interesting new question, namely, whether the HRF tailoring provided by GLMsingle might be simply due to choosing a better HRF that broadly describes the responses from a given subject (or region), or whether there is actually substantial benefit associated with voxel-wise variability in HRF shape within a given subject (or region).

To provide further insight into HRF variation, the revised manuscript now includes analysis of the stability of optimal HRF indices (see Figure 2—figure supplement 1; Results section; Methods section). We show that in active voxels, there is a clear structuring of optimal HRF indices in the form of a low-frequency spatial gradient. Moreover, the optimal indices are consistent from session to session (see panel c). This provides evidence that, beyond identifying subject-specific deviation from the assumed HRF, the FitHRF procedure may confer benefit by identifying voxel-to-voxel differences in HRF shape.

(Note: Getting into vascular issues is beyond the scope of the manuscript, but we would like to point out that spatial variation in HRF shape has a strong biophysical basis in terms of properties of the vasculature (e.g., see Kay et al., *Nature Methods*, 2020). This is mentioned in the Results section under "GLMsingle improves the reliability of beta estimates".)

It should be possible to quantify whether there is substantial dissimilarity in the chosen HRF from voxel to voxel. Since the HRFs span an equidistant arc, it would be expected that HRFs at opposite ends of the set are maximally dissimilar. Since the HRF has a biological interpretation, if it were frequently the case that neighboring voxels had dissimilar HRFs, this would be concerning.

We agree that it would be concerning if the optimal HRF indices were highly variable between neighboring voxels.

The analysis we conducted in response to Essential Revision 5.1 addresses the issue raised by the reviewers. In this analysis, we assess (quantitatively and qualitatively) the stability of the optimal HRFs in space and in time. In active voxels from a representative subject, we find that neighboring voxels rarely differ by more than a few indices, and that the optimal HRFs generally follow a low-frequency gradient over cortex (see Figure 2—figure supplement 1b). Moreover, the pattern of optimal HRFs is highly stable over the 10 experimental sessions analyzed from this subject (mean *r =* 0.78 between sessions in the most active voxels, see Figure 2—figure supplement 1c). All together, these results add evidence that the optimal HRF indices estimated using GLMsingle are far from random (though, as expected, in voxels with little or no fMRI signal, HRF indices are essentially random; see ON-OFF R2 < 10, Figure 2—figure supplement 1, top row).

6) It is a very small effect, but it would be interesting if the authors could speculate on the cost imposed by GLMdenoise in the very most-reliable voxels. Is this an artifact of the relatively small number of voxels that surpass this threshold? Or is there a chance the GLMdenoise step is removing signal? This refers to the rightmost point in the left plot of 3A (solid purple vs red; green vs orange). This is evident again in Figure 5B intra-NSD plot in the non-monotonicity from b2 to b3 for higher thresholds, and again when the b2 and b3 lines in Figure 6A (NSD) cross at r=0.25. Given that this does not seem to happen at all for BOLD5000, it is probably just an artifact, but seems nonetheless interesting enough for the authors to double-check whether there is any other explanation apparent in the data.

We thank the reviewers for pointing out this observation. Inspection of the counts of voxels included at each reliability threshold suggests that such effects are indeed an artifact of low voxel count and should not be interpreted as a loss of signal arising from the use of GLMdenoise. For example, in Figure 3A, at reliability threshold *r* = 0.5, the four NSD subjects show 318, 217, 95, and 76 voxels exceeding threshold within the nsdgeneral mask, and the four BOLD5000 subjects show 3, 1, 2, and 0 voxels respectively. (In general, there are very few voxels in BOLD5000 for which the aggregate reliability scores (averaged across β versions) reach levels that are comparable to the most reliable voxels in NSD.) As a result, plotted traces at the highest reliability levels likely contain inconsistent numbers of subjects. For all plotted results, any differences between β versions that fall within the shaded SEM regions should be interpreted with extreme caution for this reason.

As discussed above (see Essential Revisions 2–4), GLMdenoise is designed to be conservative – both in the selection of the noise pool voxels and in the choice of the number of PC regressors to include. The procedure is constrained in a manner that seeks to directly avoid the removal of valid signal from the data. Moreover, GLMdenoise as a standalone technique has been extensively validated in other work (see references to Kay et al., 2013, Charest et al., 2018).

The revised manuscript now discusses this issue and informs readers that trends in the right-most bins should be interpreted with the caveat that relatively few voxels exceed the reliability threshold at those levels. Notably, our new analyses of the auditory data from the StudyForrest dataset (see Figure 4) provide the opportunity to examine results in cases where there are very robust measures of voxel-wise reliability (given the large number of repeated trials per condition in that dataset). These results confirm that GLMsingle provides benefits even for the most reliable voxels.

7) The benefit of b4 vs. b1 seems much larger in the NSD dataset than in the BOLD5000 dataset (Figure 2A). Is this because GLMsingle was initially optimised for the NSD dataset, or is there a different reason for this? The authors mention the fact that were fewer stimulus repetitions in BOLD5000 – but isn't that exactly the scenario for which GLMsingle was intended? Could it be due to the longer ITIs in BOLD5000?

As the reviewers note, the voxel-level summary of reliability comparing b1 vs. b4 (Figure 2A) shows a more clear and consistent improvement in the NSD dataset compared to BOLD5000. There are several possible reasons for this:

1) First, a somewhat mundane reason is that BOLD5000 has many fewer instances of repeats, and this scarcity of repeats limits the robustness of the measures of reliability that we can quantify in the data. Thus, the reliability results obtained for any single voxel in BOLD5000 should be treated with caution, and we should make inferences primarily with respect to overall trends in the data.

2) Second, BOLD5000 has, in general, substantially weaker signal-to-noise than NSD. This is likely due to being collected at 3T (as opposed to NSD’s 7T) and involving a somewhat less potent experimental paradigm (brief 1-s stimulus presentation with a valence task vs. NSD’s 3-s stimulus presentation with a continuous recognition task). We suggest that, somewhat ironically, the benefits of denoising via GLMsingle are likely maximized at moderate levels of signal-to-noise. With limited signal-to-noise, it is difficult to estimate signal characteristics (e.g. HRF indices) and noise characteristics (e.g. optimal number of noise PCs), and so benefits of denoising may be modest. On the other hand, at very high signal-to-noise levels, signal quality and reliability are already high, and so there is little denoising left to be done. One way to think about it is that GLMsingle is a *data-driven* technique, and hence, the benefit of denoising may increase as signal quality increases. In fact, the results from the StudyForrest dataset directly bear this out in terms of greatest benefits being obtained for voxels with moderate signal-to-noise levels (see Figure 4c, left panel).

3) Finally, the reviewers are correct in pointing out that the longer ITIs in BOLD5000 provides yet another point of contrast with respect to NSD. The longer the ITI, the less overlap there will be in the BOLD response to successive trials, and hence the benefits of the ridge regression component of GLMsingle will be more modest compared to cases with shorter ITIs.

We have added several sentences to the Results section where Figure 2 is discussed to provide added context to aid in readers’ interpretation of the results.

8) The two datasets GLMsingle was tested on did not have a jittered intertrial interval (although the second one had a quite long (9s) intertrial interval). So, it remains to be seen whether there are also such large improvements when applying this method to a design with jittered intervals.

This is a useful observation. Many fMRI studies employ jittered intertrial intervals, and it makes sense that readers would be curious to know whether GLMsingle would confer similar benefit in such datasets compared to that observed in NSD and BOLD5000. We point out that from the perspective of statistical theory, there is no reason to think that the GLMsingle procedures are somehow specific to fixed-ITI designs. Nonetheless, it does provide value to demonstrate empirical results on additional diverse datasets.

To address the specific concern about jittered ITIs, as well as to test the general applicability of GLMsingle more broadly, we applied the GLMsingle pipeline to a third validation dataset, the music-listening component of the StudyForrest experiment. The revised manuscript provides specific details regarding the characteristics of this dataset and our analyses (see Figure 4; Results section; Methods section). In brief, the dataset measures brain responses as subjects listen to 25 musical clips of duration 6 s from 5 genres, with 8 repetitions per condition per subject. We analyze the group of 16 subjects from this dataset for whom all functional data files are available on the online data repository. The results are summarized in Figure 4. As in NSD and BOLD5000, we find that each optimization stage within GLMsingle confers added benefit to voxel reliability in the StudyForrest dataset, in a reasonably consistent manner across subjects.

9) It would also be useful to include some intermediate results for the interested reader. As an example for the two chosen dataset, it could be instructional to know how many different hrf functions were obtained using FitHRF, how the ridge regression affects shrinkage of betas etc. The provided example in the toolbox (Python / Matlab) serves well to explore some of these intermediary steps, but some of these could also be explained in further detail as supplementary material. This would have a didactic purpose, informing the reader more about the process under the hood rather than just how the choices influence final estimates of betas.

We agree that readers and users would benefit from clearer implementational guidelines and additional discussion to make the inner workings of GLMsingle more transparent.

As covered in Essential Revisions 1–2, we have extended the manuscript’s Discussion section to include more detail about the intermediate steps in the pipeline. In addition, as covered in Essential Revision 5, we have conducted new analyses that provide insight into the distribution of HRFs that are chosen in a representative subject (see Figure 2—figure supplement 1b). Due to space constraints, we suggest that the additional content related to inspection of other intermediate results is best conveyed through the example scripts on the GitHub repository.

10) Some relevant information on the amount of data from the two datasets could be explained in their Results section, specifically including number of conditions, repetitions per condition, and functional runs. It is not so straightforward to figure out this from the methods section given that the authors provide information on the datasets themselves, and then also the amount of the data used. Having some metrics in the main text would help to orient the reader and more easily allow a comparison of the results to the types of designs readers may be considering.

We thank the reviewers for pointing out that we could enhance the clarity of the Results section by providing these details.

Relevant metrics for each dataset have been added.

11) Researchers who study reliability often will complain about the use of Pearson correlation in that context. For completeness, the authors might want to at least look into this debate and decide whether it is worth addressing in the manuscript.

We recognize that there are other metrics (e.g. Spearman rank correlation) for quantifying the test-retest reliability of signal estimates (or signal-noise ratio more broadly). In the manuscript, we apply in a straightforward manner the approach of Tarhan and Konkle (*Neuroimage*, 2020), which involves correlating each voxel’s response profile over conditions between different splits of the data. We believe this approach to be well validated and reasonably standard across the field of cognitive neuroimaging. In addition, it is related closely to the metric of noise-ceiling SNR (“ncsnr”) described in detail in the NSD dataset paper (Allen et al., 2022). We anticipate that the basic trends in our results would be unchanged were we to use other metrics for the reliability of signal estimates.

We have added a sentence to the Methods section where the test-retest reliability computation is discussed to provide added context.